# D-serine mitigates cell loss associated with temporal lobe epilepsy

Stephen Beesley [1], Thomas Sullenberger[1], Kathryn Crotty [1], Roshan Ailani[1], Cameron D'Orio [1], Kimberly Evans [2], Emmanuel O. Ogunkunle [2], Michael G. Roper [2] & Sanjay S. Kumar [1]✉

Temporal lobe epilepsy (TLE) is the most common type of drug-resistant epilepsy in adults, with an unknown etiology. A hallmark of TLE is the characteristic loss of layer 3 neurons in the medial entorhinal area (MEA) that underlies seizure development. One approach to intervention is preventing loss of these neurons through better understanding of underlying pathophysiological mechanisms. Here, we show that both neurons and glia together give rise to the pathology that is mitigated by the amino acid D-serine whose levels are potentially diminished under epileptic conditions. Focal administration of D-serine to the MEA attenuates neuronal loss in this region thereby preventing epileptogenesis in an animal model of TLE. Additionally, treatment with D-serine reduces astrocyte counts in the MEA, alters their reactive status, and attenuates proliferation and/or infiltration of microglia to the region thereby curtailing the deleterious consequences of neuroinflammation. Given the paucity of compounds that reduce hyperexcitability and neuron loss, have anti-inflammatory properties, and are well tolerated by the brain, D-serine, an endogenous amino acid, offers new hope as a therapeutic agent for refractory TLE.

[1] Department of Biomedical Sciences, College of Medicine & Program in Neuroscience, Florida State University, Tallahassee, FL 32306-4300, USA. [2] Department of Chemistry and Biochemistry, Florida State University, Tallahassee, FL 32306, USA. ✉email: sanjay.kumar@med.fsu.edu

Despite recognition that temporal lobe epilepsy (TLE) is neurodegenerative in nature[1,2], mechanisms underlying loss of neurons in MEA and the sequence of events leading to spontaneous recurrent seizures or chronic epilepsy have remained largely obscure and elusive[3]. Whether neuronal loss is the cause or result of epilepsy is unclear, as are the roles of neuronal hyperexcitability and glial-mediated neuroinflammation in the epileptogenesis. The discovery of a novel class of N-methyl-D-aspartate receptors (NMDARs) in this region, which are antagonized by D-serine, has provided the opportunity to examine these issues afresh and test specific hypotheses[4,5].

Glutamatergic NMDARs are heterotetrameric proteins whose three subunits GluN1, GluN2, and GluN3 are derived from distinct gene families (Grin1–Grin3). All NMDARs contain one or more of the obligatory GluN1 subunits, which when assembled with GluN2 (A–D) subunits of the same type, give rise to conventional diheteromeric (d-) NMDARs. Triheteromeric NMDARs, by contrast, contain three different types of subunits, and include receptors that are composed of one or more subunits from each of the three gene families, designated t-NMDARs[6]. We showed previously that GluN3-containing t-NMDARs can be distinguished from GluN2-containing d-NMDARs electrophysiologically, have reduced affinity for $Mg^{2+}$, and increased selectivity for $Ca^{2+}$ over $Na^+$, making them highly $Ca^{2+}$ permeable[7]. These receptors are blocked by the pan-NMDAR antagonist D-(-)-2-Amino-5-phosphonopentanoic acid (D-AP5) and by D-serine, a potential gliotransmitter and a co-agonist of conventional NMDARs[4,6] (Supplementary Fig. 1). Given D-serine's dual role as an agonist of d-NMDARs and an antagonist of t-NMDARs, we applied it focally to the MEA to see if it mitigates TLE-related neuronal loss by preventing NMDAR excitotoxicity. The concentration of D-serine chosen (100 μM) was based on its ability to significantly suppress GluN3-containing t-NMDAR-mediated excitatory postsynaptic currents in acute brain slices[5]. In addition, we determined if exogenous application of D-serine counters the epilepsy-mediated effects on astrocytes (reactive status and numbers) and curtails proliferation and/or infiltration of microglia into the MEA, thereby suppressing neuroinflammation.

To this end, we stereotaxically implanted adult rats (postnatal days 40–45) with cannulae allowing precise unilateral access to the right MEA (non-cannulated animals served as controls; Fig. 1a). Cannulated and non-cannulated animals were then subjected to an intraperitoneal injection of the chemoconvulsant pilocarpine to induce status epilepticus, which serves as the initial precipitating injury or insult (see description of the pilocarpine model of TLE[8] in Methods). Following recovery from status epilepticus (terminated with diazepam), rats are either administered artificial cerebrospinal fluid (aCSF, vehicle) or D-serine via cannula daily for ~4 weeks (latent period) during which they are video-monitored for frank epileptic seizures. Seizure severity is scored on a modified Racine scale[9] (see inset in Fig. 2a). D-serine was also administered continuously in a separate cohort of animals using mini osmotic pumps surgically implanted under the scruff. The latent period following initial insult coincided with the maximum length of time for which D-serine could be infused continuously. Animals that survive the pilocarpine treatment are deemed epileptic if they have three or more distinct episodes of spontaneous seizure activity (severity 3 or above on the Racine scale) and nonepileptic if no seizure activity is observed post status. Animals having fewer than three seizures were classified as pre-epileptic. Non-status animals in which pilocarpine failed to induce status epilepticus were found to be seizure-free throughout the period of observation.

## Results

### D-serine is effective in averting epileptogenesis. Daily infusion of D-serine lowered the percentage of post-status animals (100%)

from becoming epileptic (25%) compared with vehicle infused (73%) or non-cannulated (84%) cohorts (flagged yellow near the bottom of Fig. 1b; Supplementary Fig. 2). Interestingly, continuous infusion of D-serine (100 μM at 0.1 μl/h) countered the chemoconvulsive effects of pilocarpine preventing a significant number of animals from developing status epilepticus (61%) compared with the daily-infused aCSF (38%), D-serine (20%), or non-cannulated (25%) cohorts (flagged green near the top of Fig. 1b). These data demonstrate D-serine's role in averting epileptogenesis. Furthermore, focal application of D-serine to MEA proved anticonvulsive[10], which is counter to its presumed role as a NMDAR agonist and proconvulsive[11].

To correlate behaviorally observed changes in epileptogenic status of animals with changes in neuron-glia pathology in MEA, under conditions of daily infusion of D-serine or aCSF, we harvested the brains of animals at the end of the video monitoring period, 1-month after initial insult. A small percentage of these animals were lost during tissue processing and were omitted from further analysis. The revised estimates indicate that ~82% of post-status animals infused with aCSF became epileptic compared to 27% with D-serine; 9% of post-status animals infused with aCSF were rendered nonepileptic compared to 46% with D-serine and 9% of post-status animals infused with aCSF were deemed pre-epileptic compared to 27% with D-serine (Fig. 2a). Non-status animals receiving either D-serine or aCSF were normal in every aspect and could be distinguished from post-status animals based on the pattern of weight gain following initial insult (Fig. 2b). D-serine had two notable effects in post-status animals; first, it prolonged the time to first seizure from ~9 days in aCSF to ~17 days (Fig. 2c) and second, it reduced seizure severity from $4.2 \pm 0.1$ (mean ± SEM) in aCSF to $3.6 \pm 0.1$ measured on the Racine scale ($p < 0.002$, t-test; Fig. 2d). All brains harvested intact from animals were cut into 50 μm-thick horizontal sections and one-in-six series per animal, covering the dorsal-ventral extent of MEA, was used for histology and cell counting (Fig. 2e).

### D-serine treatment attenuates TLE-mediated neuron loss. Nissl staining with thionin (1:100 of 0.25% stock solution) revealed a conspicuous loss of neurons throughout layer 3 of the MEA in aCSF-infused post-status animals compared with non-status controls. Post-status animals treated with D-serine resembled controls with only a moderate loss of neurons (Fig. 2f). Interestingly, D-serine mediated rescue of neurons was observed bilaterally despite its unilateral infusion into the right hemisphere. The mechanisms underlying this lateralization are unknown but likely include ventricular transport and/or cross hemispheric connectivity of neuron between the left and right MEAs[12–14]. To estimate the number of Nissl-stained neurons in layer 3 of MEA under various conditions, we undertook a detailed stereological analysis using wet-mount sections, which enabled finer resolution of overlapping neurons yielding better estimates (Fig. 2g). Indeed, there were far fewer neurons in the MEA of aCSF-infused post-status animals compared with aCSF or D-serine-infused non-status controls in both hemispheres (Fig. 2h). Averaged neuron counts were reduced from $108 \pm 4$ thousand (aCSF) and $110 \pm 3$ thousand (D-serine) in non-status animals to $70 \pm 3$ thousand in aCSF-infused post-status animals, an overall reduction of ~35% rendering 82% of animals epileptic. Treatment with D-serine significantly attenuated neuron loss to $87 \pm 5$ thousand, a rescue of ~15% of all neurons in MEA which was adequate to lowering the percentage of animals becoming epileptic from 85 to 27% and increasing the percentage of nonepileptic animals from 9 to 46% (Figs. 2h, a). Neuron counts were on average greater in post-status animals treated with D-serine than with aCSF irrespective of whether they became epileptic or pre-epileptic (Fig. 2h,

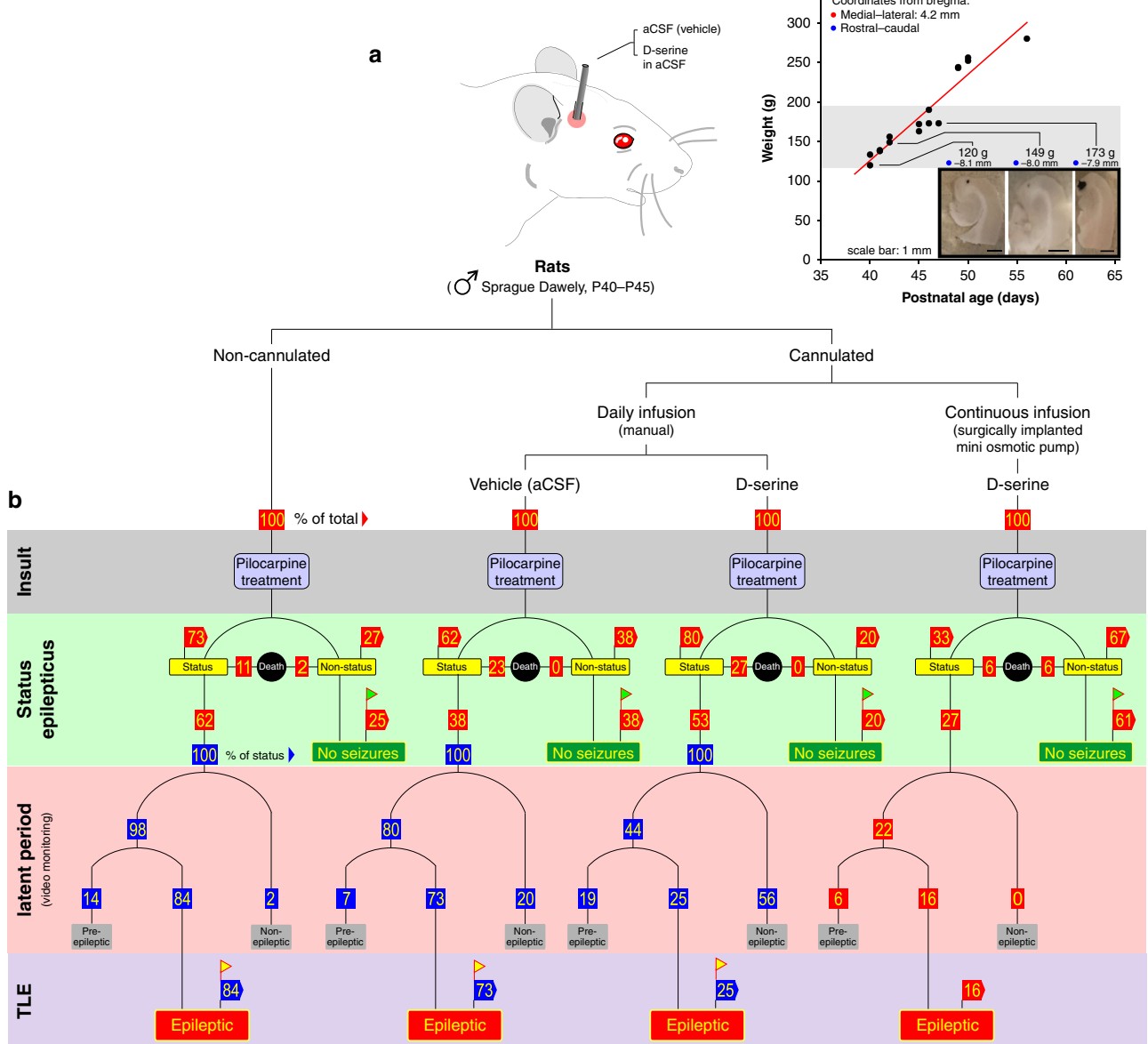

**Fig. 1 Assaying D-serine's efficacy in preventing temporal lobe epilepsy. a** Schematic and details of the route and mode of D-serine administration. Note changes in stereotaxic coordinates required to accurately target the right MEA with a cannula (confirmed through injection of India ink, *inset*) as a function of age and body weight of the rat (gray area indicates range of animals used). The non-cannulated cohort of animals includes those utilized in past studies of TLE in the laboratory[32,33]. The MEA is also referred to as area entorhinalis pars medialis in the literature[34]. **b** Tree diagram showing behaviorally observed outcomes (status epilepticus, frank seizures, epilepsy or death) of pilocarpine treatment (insult) under conditions identified in **a**. The boxed numbers indicate percent of total animals (red) or percent of animals with status epilepticus (status, blue), under various regimens (changes are flagged to encourage comparison). All video recordings of behavior are available upon request.

rightmost panel). In the few cases where post-status animals were rendered nonepileptic, neuron counts with D-serine were like those with aCSF. Neuron loss in layer 3 of MEA is therefore interlinked with epileptogenesis because D-serine-mediated rescue of these neurons prevented animals from becoming epileptic and continuous infusion of D-serine countered the chemoconvulsive effects of pilocarpine preventing them from developing status epilepticus. We did not distinguish between excitatory and inhibitory neurons in this study, given that both neuron types have been shown to perish in TLE[15]. Given that D-serine rescue of neurons was observed bilaterally despite its unilateral infusion suggests that it somehow gets to the uninfused hemisphere internally. Considering that neuron counts were similar in both hemispheres after D-serine despite the potential for dilution in

crossing hemispheres implies that it could even be efficacious at doses smaller than the one used in this study.

**D-serine mediated rescue of neurons is inversely correlated with astrogliosis and microgliosis.** Astrocytes, like neurons, are a putative source of D-serine in the brain (express serine racemase, the enzyme that converts L- to D-serine)[16,17] and therefore capable of regulating endogenous levels within MEA. Hence, we asked if their numbers and/or reactive status were also affected by TLE[18]. We discovered a sparse distribution of astrocytes interspersed among neurons in the MEA of non-status controls (including astrocytic sheathed vasculature) (Fig. 3a). This pattern was strikingly altered in the epileptic animals and it appeared as

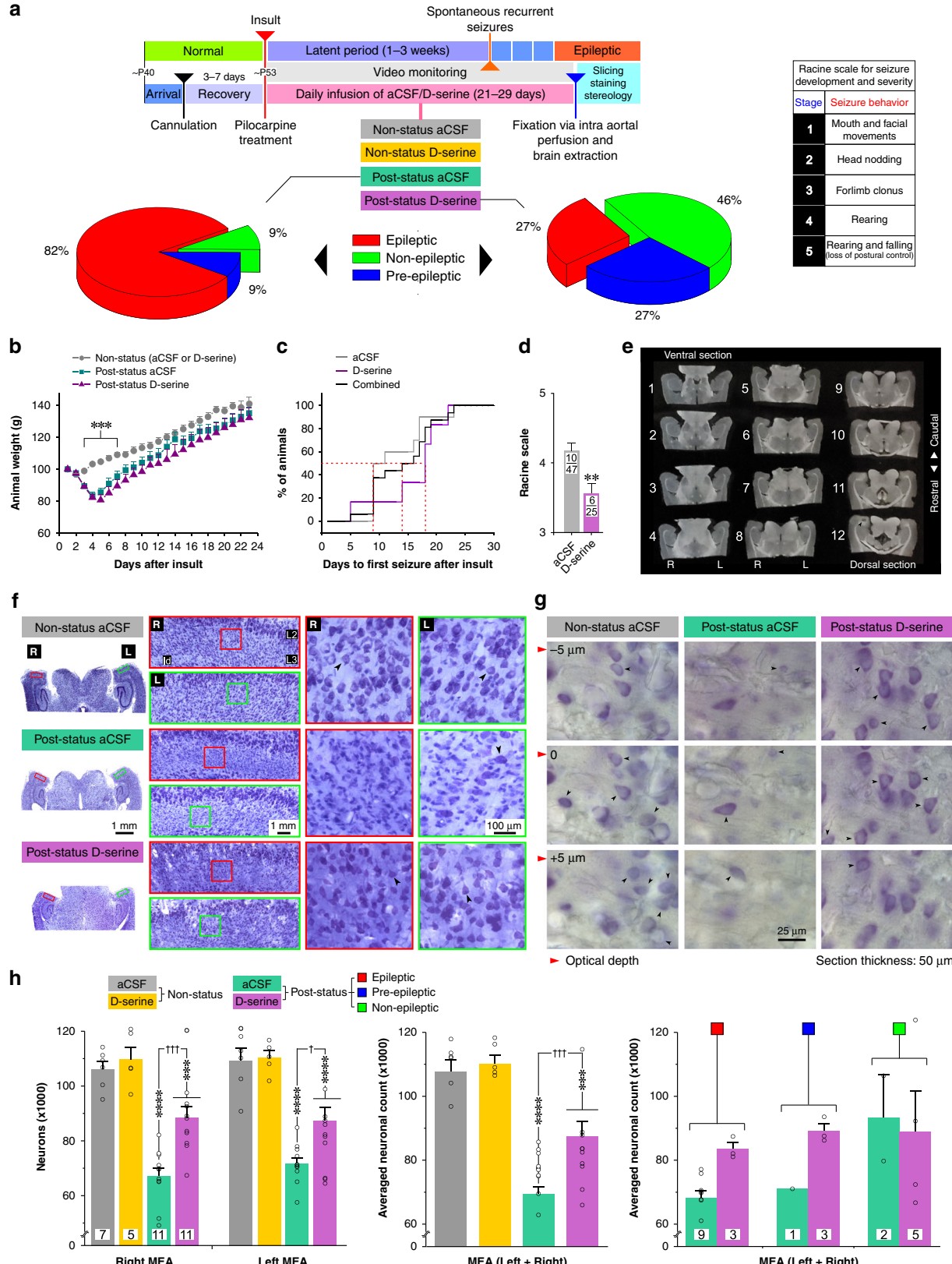

though the entire layer 3 neuronal population was replaced by astrocytes (top panels, Fig. 3a). This upregulation of astrocytic signal in the MEA was curtailed to a significant extent by treating post-status animals with D-serine and confined to just the midmost portions of the MEA (middle panels, Fig. 3a). The few animals in which D-serine intervention was unsuccessful, possibly on account of the tight tolerances to gain access to the region when placing the cannulae, also showed expansion of the epileptic focus laterally into the rest of the MEA starting from this location. Astrogliosis was manifest bilaterally, although there was a small but significant increase in astrocyte counts on the side that was cannulated versus the contralateral side in epileptic rats

**Fig. 2 D-serine reduces number and severity of frank seizures and attenuates neuronal loss in the MEA. a** Timeline of experimental manipulations and behavioral outcomes (color coded). Pie charts indicate percent of post-status animals infused with aCSF (vehicle, left) or D-serine (right) that were categorized, based on video monitoring, as nonepileptic, pre-epileptic, or epileptic at the end of the latent period (subsequently used for stereological counting of neurons within MEA). Inset, the modified Racine scale used in this study for assessing seizure development and scoring severity. **b** Profile of changes in animal weight following initial insult ($n = 15, 14, 11$ animals for non-status aCSF or D-serine, post-status aCSF, and post-status D-serine, respectively). Error bars represent SEM. ***$p < 0.001$, two-way ANOVA with a Tukey's post-hoc correction. **c** Cumulative distribution functions for changes in days to first seizure following initial insult with and without D-serine on board. **d** Severity of spontaneous seizures scored on a modified Racine scale (3–5) for animals infused with aCSF or D-serine. The number of animals used ($n$, numerator) is delineated from the total number of seizures recorded in them (denominator) within the bar plots. Error bars represent SEM. **$p < 0.01$, Student's $t$ test. **e** An example of a complete one-in-six series of brain sections (50 μm thick, covering the dorsal-ventral extent of the MEA; R right, L left) used for stereological counting of neurons. Note the ventral most location of the cannula (black arrowhead, section no. 12). **f** Brain sections from control and post-status rats treated D-serine or aCSF (vehicle), stained with thionin, and viewed at different magnifications (each set of panels, from L to R, are zoomed images of boxed regions, red (right), green (left), in the previous set). Note the conspicuous absence of Nissl-stained neurons (arrowheads) in layer (L) 3, but not L2, of MEA in aCSF-infused post-status rats. ld is lamina dissecans. **g** Images of sampling sites (at an optical depth interval of 5 μm) in wet-mounted brain sections from animals in various groups used for stereological counting of neurons. Neurons were counted based on visualization of their nucleoli (arrowheads). **h** Neuron counts estimated using stereology for non-status (controls) and post-status rats treated with D-serine or aCSF (left panel). Raw data and histograms of averaged neuronal counts from both hemispheres (middle panel) and a breakdown based on epileptogenic status (right panel; data presented in the interest of full disclosure and not used for any statistical comparison). The total number of animals used in this part of the study are indicated in the bar plot to the left of the panel). The numbers in the bars indicate the total number of animals ($n$) used. ***$p < 0.001$, ****$p < 0.0001$, one-way ANOVA with a Dunnett's post-hoc correction. Error bars represent SEM.

(Supplementary Fig. 3). These data demonstrate that neuronal loss in the MEA is correlated with the observed astrogliosis.

To characterize the glial pathology further, we counted all GFAP-positive cells (astrocytes; Fig. 3b) and CD11b/c (OX42)-positive cells (microglia, Figs. 3c and 4a, b) in layers 2 and 3 of MEA in a subset of sequential sections (5–8, Fig. 2e) from animals in various stages of epileptogenesis with or without D-serine on board. The density of astrocytes in MEA was two-fold higher in aCSF-infused post-status rats compared with non-status controls ($849 \pm 41$ versus $423 \pm 41$ cells per mm$^2$, $p < 0.001$, Fig. 3b) and treatment with D-serine, significantly reduced their numbers compared with the epileptics ($617 \pm 36$ cells per mm$^2$, $p < 0.001$). A similar albeit more dramatic trend was also observed with microglia that were essentially absent in control animals but were expressed copiously in the epileptics (control: $23 \pm 8$ versus epileptic: $428 \pm 37$ cells per mm$^2$, $p < 0.001$, Figs. 3c and 4a, b). As with astrocytes, treatment with D-serine, significantly reduced their numbers in the MEA compared with the epileptics ($212 \pm 46$ cells per mm$^2$, $p < 0.005$). Furthermore, changes in astrocyte and microglia counts were manifest mainly in layer 3 but not in layer 2 of the MEA, although we did note a small but nonsignificant increase in microglia in layer 2 as well in epileptic animals.

To gauge reactive status of astrocytes (not possible for microglia), we assayed morphometric changes in their arborization and total volumes under epileptic and nonepileptic conditions using Sholl analysis[19] (Fig. 3d). We found a significant increase in astrocytic volume in MEA of epileptic animals compared with the nonepileptic controls (control: $35 \pm 6$ versus epileptic: $91 \pm 11$ μm$^3$, $p < 0.001$, Fig. 3e), despite a similar degree of arborization[20] (Fig. 3f). Treatment with D-serine curtailed volumetric expansion in astrocytes ($53 \pm 5$ μm$^3$, $p < 0.01$); however, the plot of the number of crossings as a function of radial distance was now shorter and left-shifted, indicating diminished arborization. We believe that astrocytes react differently to reductions in ambient D-serine levels, as is likely in TLE, than they do to its replenishment, as suggested by their segregation into distinct nonoverlapping regions on the volume versus crossings plot (Fig. 4g). This is not surprising, given that astrocytes have been proposed to exhibit multiple distinct activation states based on ambient conditions[21].

**Neuronal loss precedes neuroinflammation in the MEA.** Microglia, essentially absent in the non-status controls, were

strikingly manifest in epileptic animals (top panels, Fig. 4a) and D-serine greatly reduced their proliferation and/or infiltration into MEA (bottom panels, Fig. 4a). Given the similarities in microglial density between D-serine-treated nonepileptic animals (partial loss of neurons), and the few that became epileptic despite it being onboard (significant loss of neurons), suggests that microglia per se do not bring about epilepsy. The quadruple-stained sections (include the nuclear counterstain DAPI, Fig. 4b) showcase TLE-related pathology and the overall effects of D-serine intervention from the perspective of all three cell types — neurons, astrocytes, and microglia. To determine if alterations in their immunoreactivity correlated with their counts under various epileptogenic and nonepileptogenic conditions, and to establish whether loss of neurons triggers neuroinflammation or vice versa, we measured the relative abundance and expression profiles of NeuN (neurons), GFAP (astrocytes), and Iba1 protein (microglia) in the MEA of post-status animals at various time points following initial insult (Fig. 4c). We used tissue from the MEA of non-status animals and the cerebellum as controls for our western blot analysis (Fig. 4d). Our data show that neuronal loss is manifest as early as 1 day following initial insult in post-status animals compared with the non-status controls (upper panel, Fig. 4e). Although NeuN expression recovers somewhat by the end of the latent period (~29 days), it stabilizes below pre-insult levels, indicating permanent loss of neurons in the MEA. The changes observed were specific to the MEA and not seen in the cerebellum. Unlike neurons, astrocytes and microglia were not manifest in the MEA until post insult day 5 at which point there was a marked increase in their relative abundance. Interestingly, while the astrocytic levels stayed elevated by the end of the latent period, microglia levels tended to decay back to baseline (lower panels, Fig. 4e). The temporal sequence of these events suggests that astrocyte and microglia mediated neuroinflammation in TLE is triggered by the loss of neurons within the MEA (and not vice versa), and by the time to first seizure following initial insult (approximately day 10, Fig. 2c), their numbers are at or near their peak values.

**Neurons and astrocytes are both sources of D-serine in the MEA.** To locate the source of endogenous D-serine in the MEA, we assayed for the expression of serine racemase, the enzyme responsible for the synthesis of D-serine from L-serine, in neurons and astrocytes in epileptic and control tissue (Figs. 5a, b;

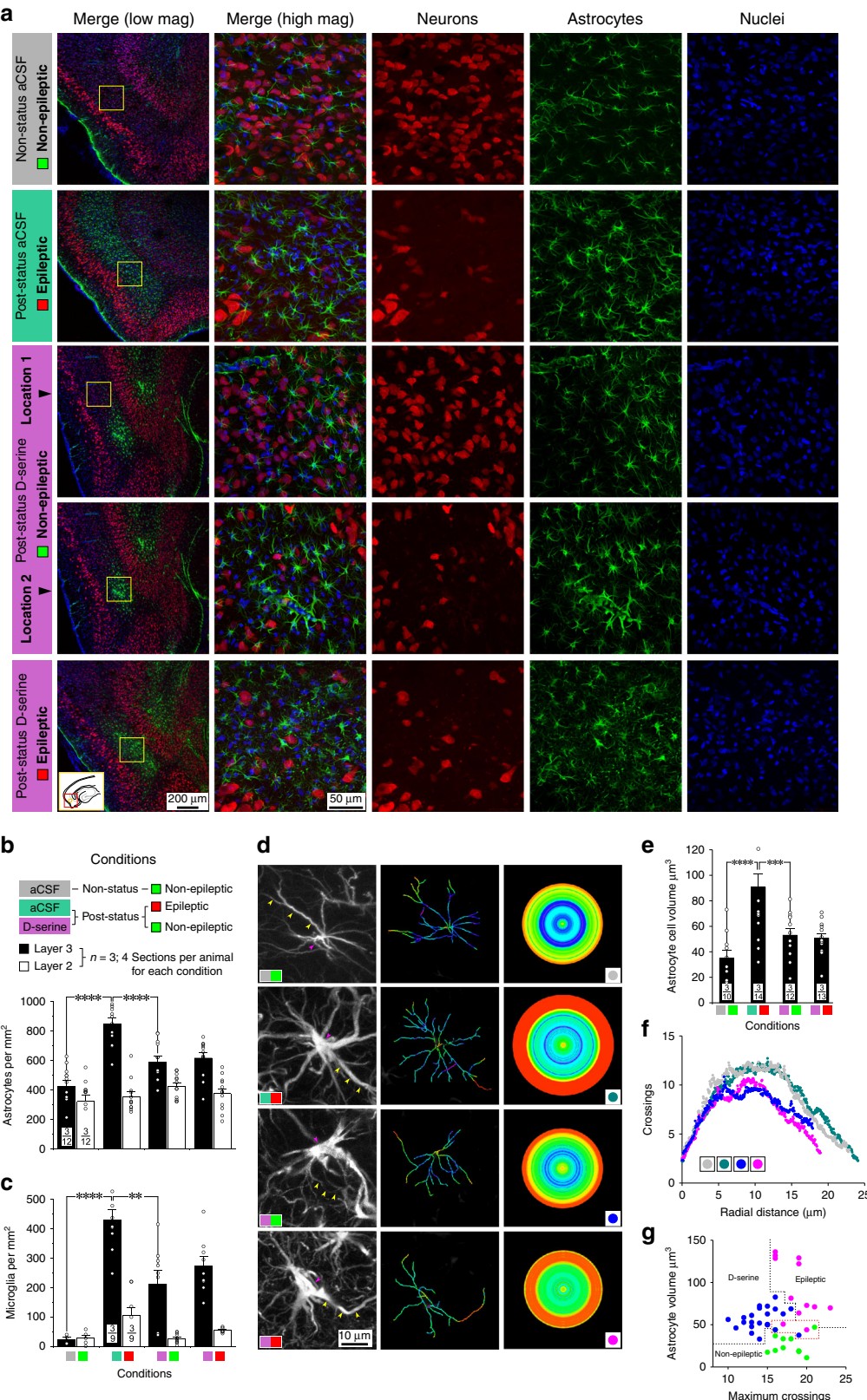

$n = 4$ sections from two animals in each group). Immunohis-
tochemistry with fluorescently tagged antibodies against serine
racemase, NeuN and GFAP showed clear colocalization with both
neurons and astrocytes in epileptic (Fig. 5a, row 4, columns 3)
and control tissue (Fig. 5b, row 2, columns 2). The specificity of
the antibody for serine racemase could be gauged by the fact that

not all neurons in the MEA were positive for serine racemase (red
arrows Fig.5a, row 5, columns 1–3) and the secondary-only
control (anti-biotin and streptavidin Alexa-594) showed no
immunoreactivity or colocalization (Fig. 5c). We are unsure
whether the racemase-negative neurons are GABAergic inter-
neurons (Fig. 5a, row 5, column 1) or a special cohort of

**Fig. 3 Focal application of D-serine minimizes astrocytic density in layer 3 of the MEA. a** Immunofluorescence images of the MEA (inset, bottom left panel) in non-status (controls) and post-status rats treated with D-serine or aCSF (vehicle). Neurons immunoassayed with fluorescently tagged antibodies against NeuN (red), astrocytes with antibodies against GFAP (green), and nuclei with DAPI (blue) shown merged (leftmost panels) and separately, as high magnification images of the boxed areas in panels to the left (yellow). Raw data and histograms of astrocytic (**b**) and microglia (**c**, immunoassayed with the anti-CD11b antibody, OX42, and shown in Fig. 4a) densities in layers 2 and 3 of MEA in animals under the conditions indicated (color codes show segregation of cohorts based on treatment regimen and final outcomes). Data within bar plots indicates number of animals used (numerator) and the total number of sections assayed for each condition (denominator). Astrocyte dendritic complexity (quantified using Sholl analysis **d**, **f**) and volume (**e**) in animals under the conditions indicated. Note differences in dendritic morphology of representative cells (left; arrowheads in **d** point to somata, purple and dendrites, yellow) and their tracings (middle), color coded for changes in the number of branches at concentric circles of increasing radii (right). Raw data and histogram of astrocytic cell volumes (**e**) and plot of the number of crossings as a function of radial distance (**f**), averaged across 10–14 cells from three animals in each cohort. **g** Scatter plot of volume versus maximum crossings for astrocytes in epileptic (pink), nonepileptic (green), and D-serine infused animals (blue). ***$p < 0.001$, ****$p < 0.0001$, one-way ANOVA with a Dunnett's post-hoc correction. Error bars represent SEM.

excitatory neurons in the MEA (Fig. 5a, row 5, column 3). Note the dense plexus of astrocytic processes in epileptic tissue spanning the entire MEA (Fig. 5a, row 4, column 1), including vasculature (bv, blood vessels; Fig. 5a, row 6, column 1), all positive for serine racemase. These data suggest that both neurons and astrocytes can be sources of D-serine in the MEA.

**D-serine levels are depleted in the epileptic brain.** Finally, to confirm whether D-serine in MEA is indeed depleted under epileptic conditions[22], we measured its abundance in brain tissue directly using chiral micellar electrokinetic chromatography (MEKC), a sensitive technique with a low limit of detection (<10 nM) that provided adequate resolution of the L and D enantiomers[23] (peaks 1 and 2 on the electropherograms, Fig. 6a; see Supplementary Fig. 4 for further validation). D-serine levels were assayed in terms of their absolute values (μg/g of brain tissue) and as percentages of total serine (D + L) in tissue samples from MEA, lateral entorhinal area (LEA) and the cerebellum of non-status control and epileptic rats following a 30 min incubation in HEPES-buffered aCSF (32 °C) after resection. Both the supernatant, containing the secreted fraction, and the cell homogenate fraction were analyzed using MEKC (Fig. 6b). We found that D-serine in MEA of epileptic animals was significantly reduced compared to controls in the supernatant (control: 1.3 ± 0.1 μg/g, 18% ± 0.9 versus epileptic: 0.8 ± 0.1 μg/g, 14% ± 0.7; $p < 0.05$ and 0.005, respectively), but not the homogenates (control: 2.7 ± 0.4 μg/g, 17% ± 0.5 versus epileptic: 2 ± 0.5 μg/g, 18% ± 2.3; $p = 0.27$ and 0.93, respectively). Furthermore, the epilepsy-related changes in D-serine levels were specific to MEA and not observed in either LEA or the cerebellum although cerebellar tissue had significantly lower levels of D-serine compared with either LEA or MEA. The observed reduction in D-serine levels in MEA correlate well with those observed in the cortex and hippocampi of post-status rats[22] and suggest that while total D-serine may be similar under control and epileptic conditions, ambient D-serine in the extracellular compartments of the MEA is significantly depleted in TLE.

## Discussion

Based on these data and the likely source of D-serine (neuronal and/or astrocytic)[24,25], we propose two scenarios explaining why D-serine levels in the MEA might be depleted in TLE and how a quick restoration or pre-treatment with it might be of therapeutic or prophylactic value (summary, Fig. 6c). The neuronal perspective — as an antagonist of GluN3-containing t-NMDARs that are expressed copiously in MEA, D-serine likely provides neuroprotection by preventing $Ca^{2+}$-induced excitotoxicity brought about through these highly calcium-permeable receptors[7]. Indeed, even within the same sections, D-serine's efficacy appears to be graded with the midmost portions of MEA where expression of these receptors is sparse and seeing the greatest damage as

neurons in this region are rendered more vulnerable. Furthermore, the prophylaxis provided by continuous infusion of D-serine into MEA against development of status epilepticus (and TLE) suggests that it acts directly by antagonizing these receptors. This might be of relevance given the possibility that neuronal hyperexcitability shuts-off presynaptic release of D-serine thereby activating postsynaptic t-NMDARs[26]. Assuming astrocytes to be the source of D-serine[27], we would expect its availability to be greater in the MEA under epileptic conditions owing to a surge in their numbers. This, however, does not appear to be the case. The glial perspective — given that normal homeostatic functions of astrocytes are affected by injury, reflected by changes in their reactive status, we propose the hypothesis of a switch in their neuroprotective role whereby they are no longer a source of D/L-serine in the MEA during TLE. Indeed, L-serine may be converted by the enzyme serine racemase to pyruvate (via a β-elimination pathway) instead of D-serine (via a racemization pathway) in reactive astrocytes to meet the high metabolic demand of attending to dead and dying cells during seizure activity and/or in neurons attending to their hyperexcitable state. Microglia can also be a source of D-serine in the MEA; however, they are only manifest in epileptic animals suggesting a minimal contribution to epileptogenesis[28,29].

The TLE pathology described here from the neuron-glia perspective finds common ground in D-serine, which might influence multiple cellular processes to curtail seizure activity, mitigate neuron loss, and prevent astrocyte and microglia-initiated neuroinflammation during the latent period. Indeed, electrographic limbic seizures and most of the other synchronous epileptiform activities seen in vitro depend on ionotropic glutamatergic transmission and on NMDA receptor-mediated mechanisms in particular, which play an essential role in sustaining ictal discharges[30]. We believe that neurodegeneration and gliosis work hand in glove to bring about the pathology in the MEA, and in the long run, feed into each other in a vicious cycle, disrupting D-serine availability and homeostasis to bring about neuroinflammation. Thus, the initial hyperexcitability induced loss of neurons might serve as a trigger for a cascade of events leading up to this state. The rescue of MEA neurons, which would otherwise perish due to t-NMDAR-mediated excitotoxicity under these conditions by D-serine, likely preserves the fidelity of excitatory and inhibitory cells and circuits against seizure activity not only in entorhinal cortex but also in downstream structures including the hippocampus thereby averting its generalization leading to full-fledged TLE. The findings outlined here are of relevance to our quest for novel therapeutic interventional strategies for combating TLE in humans.

## Methods

**Cannulation surgery.** All experiments were carried out in accordance with the National Institutes of Health Guide for Care and Use of Laboratory Animals and were approved by the Florida State University Institutional Animal Care

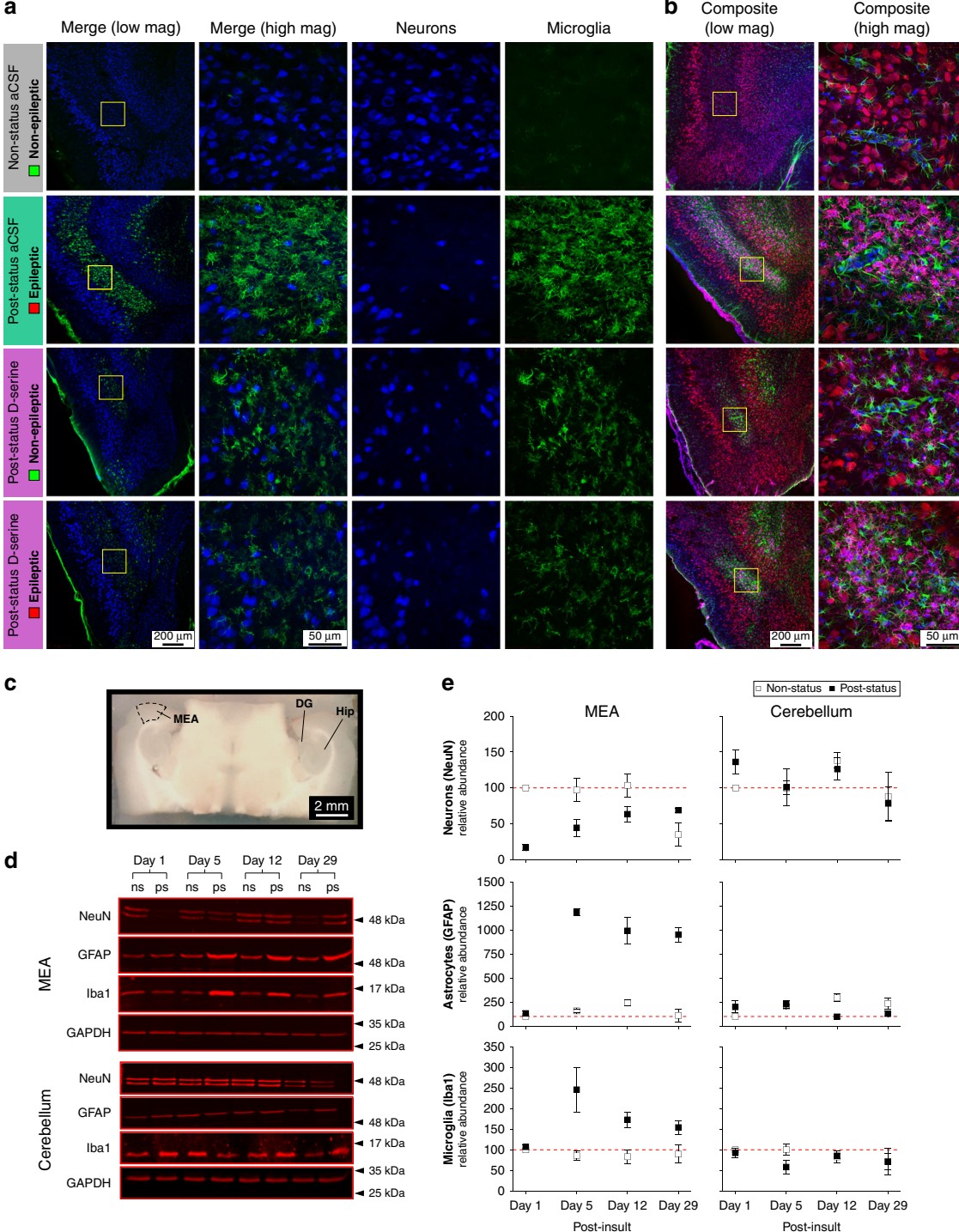

**Fig. 4 Focal application of D-serine limits microglia proliferation and/or infiltration into layer 3 of the MEA. a** Immunofluorescence images of the MEA in non-status (controls) and post-status rats treated with D-serine or aCSF (vehicle). Neurons immunoassayed with fluorescently tagged antibodies against NeuN (blue) and microglia with the anti-CD11b antibody, OX42 (green), shown merged (leftmost panels) and separately as high magnification images of the boxed areas in panels to the left (yellow). **b** Quadruple immunostaining of neurons (blue), astrocytes (green), microglia (magenta), and nuclei (blue) highlighting complete neuroglia pathology within MEA of post-status rats treated with D-serine or aCSF (panels on right are high magnification images of the boxed regions in panels on left). **c** Section of the brain slice containing the MEA that was micro dissected out for use in immunoblotting (DG dentate gurus, Hip hippocampus). Roughly four sections (600 μm thick) per hemisphere were harvested from each animal and pooled. Cerebellar tissue from these brains was used as control. **d** Representative immunoblots for NeuN (neurons), GFAP (astrocytes), and Iba1 (microglia) for MEA and cerebellum harvested from non-status (ns) and post-status (ps) rats 1, 5, 12, and 29 days post insult. GAPDH was used as loading control. Relative positions of standard molecular weight markers indicated to the right of each panel (for full scans see Supplementary Fig. 5). **e** Time course of changes in relative abundance of NeuN (neurons), GFAP (astrocytes), and Iba1 (microglia) in MEA and cerebellum quantified from the immunoblots ($n = 3$). Error bars represent SEM.

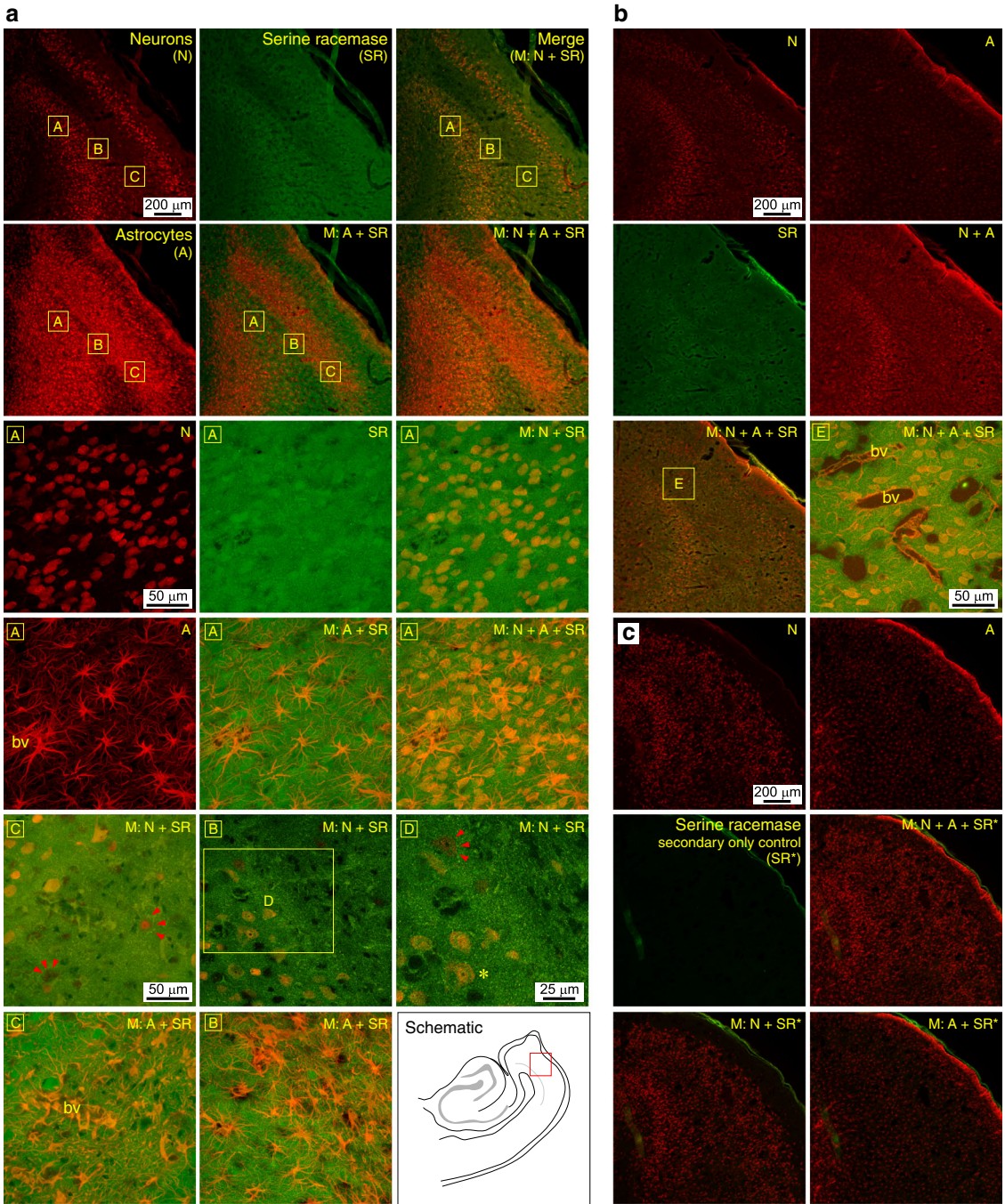

**Fig. 5 Serine racemase is localized in both neurons and astrocytes in the MEA. a** Representative images of epileptic tissue acquired on a confocal microscope showing anti-NeuN (N), anti-serine racemase (SR), and anti-GFAP (A) immunoreactivity in neurons and astrocytes within different regions of the MEA (see schematic, row 6, column 3). Fluorophores for N (red), A (red), and SR (green) have been pseudo colored to show colocalization (yellow) in the merged (M) images. Panels in rows 3 through 6 are high magnification images of the lettered boxed regions identified above or within the same row. Text (top-right) in the high magnification panels identifies the immunoreactivity shown. For example, expression of serine racemase in neurons (box A) is shown in high mag through panels in row 3 (columns 1–3) and in astrocytes through panels in row 4 (columns 1–2). Panels in row 5 showcase racemase-positive (*) and racemase-negative neurons (red arrows) within the same section. bv blood vessels. **b** Representative images from control tissue showing colocalization of serine racemase with neurons and astrocytes. The panel in row 3, column 2 is a composite high mag image of N, A, and SR in the boxed region E (row 3, column 1). **c** A secondary-only control for SR with N and A (in control tissue) highlighting the specificity of the antibody used. In the absence of the SR antibody there is neither a signal nor colocalization with N or A. Each experiment shown in **a**–**c** was repeated at least twice independently with similar results.

Committee. Sprague-Dawley rats (male, 160–190 g) were anesthetized with iso-flurane (4%, Henry Schein, Melville, New York), prepped for surgery, and secured into place in a stereotaxic instrument with the aid of ear bars. Anesthesia was maintained with isoflurane (2.5%) during surgery. The surgical site was sterilized with betadine and EtOH (70%) and an incision made down the midline to expose

the skull. Depending on the age and weight of the animal (Fig. 1a), a hole was drilled between −7.9 and −8.1 mm along the rostral-caudal axis, and 4.2 mm along the medial-lateral axis relative to Bregma, to access the right MEA with a cannula. The precise coordinates were confirmed through pilot studies of dye injection. Four additional holes were drilled to place bone screws (Fine Science Tools, Foster

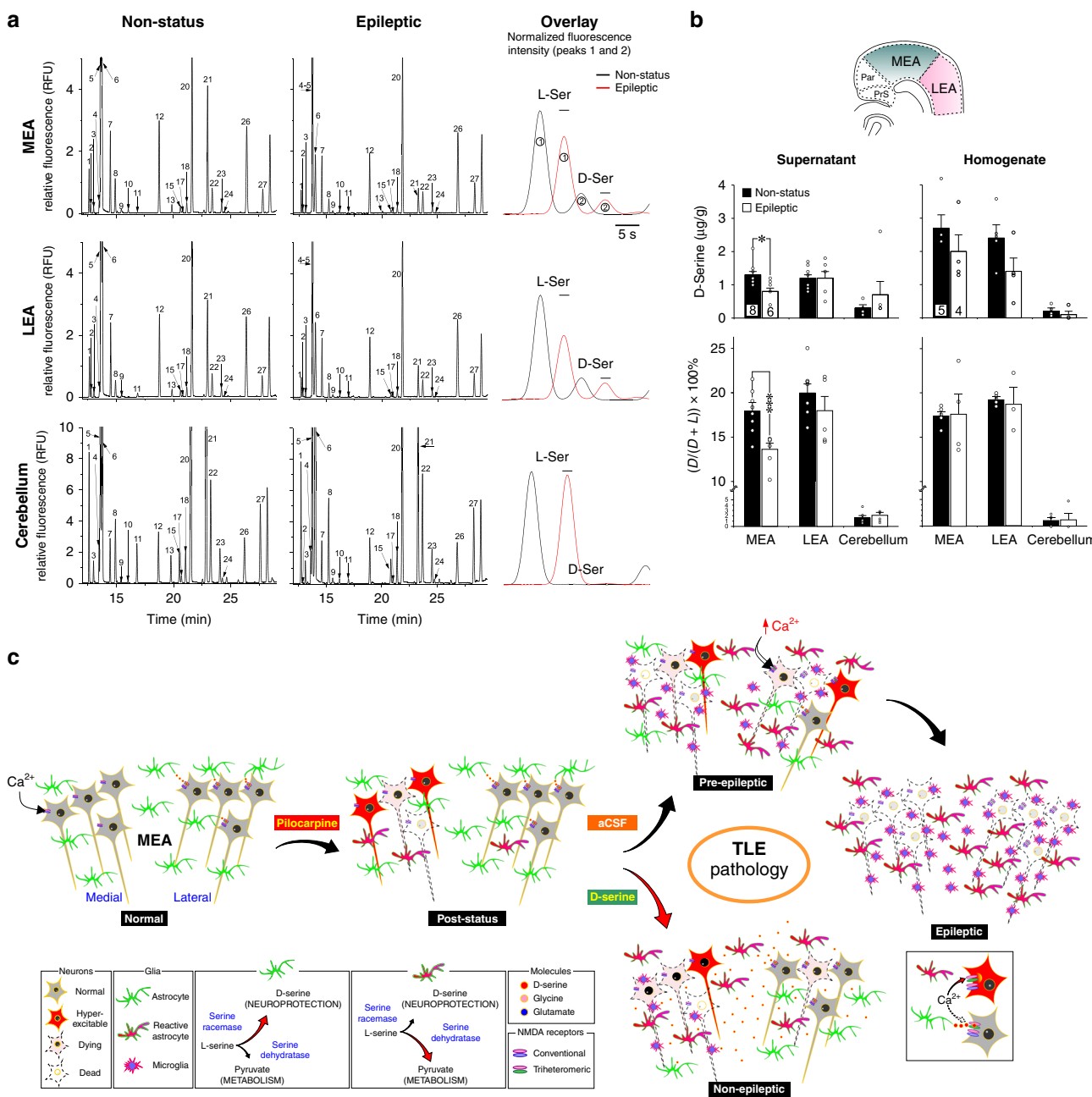

**Fig. 6 Assessment of D-serine levels in MEA and summary of neuroglia changes in TLE. a** Representative electropherograms of the secreted fractions from MEA (top row), LEA (middle row), and the cerebellum (bottom row) of non-status control and epileptic rats. Peak identification: (1) L-Ser; (2) D-Ser; (3) L-Asn; (4) L-Thr; (5) L-Gln; (6) L/D-Glu + D-Gln + D-Thr; (7) β-HSer (internal standard); (8) L/D-Asp; (9) L-His; (10) Gly; (11) L/D-Ala; (12) α-ABA (IS); (13) L-Tyr; (14) D-Tyr; (15) L-Met; (16) D-Met; (17) β-Ala; (18) L-Val; (19) D-Val; (20) Tau; (21) GABA; (22) L-Ile + L/D-Leu; (23) L/D-Phe; (24) L-Trp; (25) D-Trp; (26) β-Phe (IS); and (27) L/D-Arg. Insets (rightmost panel) are overlays of zoomed-in images of electropherograms from control (black) and epileptic (red) animals highlighting differences in the normalized fluorescence intensities of the L- and D-serine peaks (time-shifted to facilitate viewing). The bars atop peaks facilitate viewing of the amplitude differences. **b** Raw data and histograms of average D-serine levels in brain tissue (supernatant or secreted fraction, left panel; cell homogenate fraction, right panel) from the indicated regions of non-status control and epileptic rats expressed in μg/g of brain tissue (top panels) and as a percentage of total serine (D + L, bottom panels) within the samples. Numbers within bar plots indicate animals used ($n$) and the inset identifies location of LEA, presubiculum (PrS), and parasubiculum (Par) relative to the MEA. *$p < 0.05$, ***$p < 0.001$, Student's $t$ test. Error bars represent SEM. **c** Summary of neuroglia changes characterizing TLE pathology and two possible avenues through which D-serine might mitigate cell loss within the MEA.

City, CA). A custom-made guide cannula (PlasticsOne, Roanoke, VA), cut to a length of 3.2 mm, was lowered carefully and fixed in place with dental cement reinforced with bone screws. An internal cannula inserted through the guide cannula was made to extend 1 mm in depth beyond it such that the site of administration was precisely 4.2 mm ventral to the skull surface. Following placement of the cannulae and drying of the cement, the skin was pulled back together with dissolvable sutures to secure the wound. For continuous infusion of D-serine

(100 μM), a mini osmotic pump (Alzet model 1004, Durect Corporation, CA) capable of delivering at a constant flow rate (0.1 μl per hour), with a 28-day reserve, was chosen. The pump was preloaded with D-serine dissolved in aCSF and primed in bacteriostatic 0.9% saline for 48 h at 37 °C as per manufacturer's instructions. Implantation surgery was as described for placing the cannulae. The main body of the pump was placed in a pocket between skin and muscle of the scruff via an incision that was closed with dissolvable sutures along with the cranial incision.

Antibacterial gel was applied to the wound edges and meloxicam (5 mg/ml; Henry Schein, Melville, New York) was given intraperitoneally (1 mg/kg) to reduce inflammation. Saline (0.9%; 1 ml/100 g) was also administered subcutaneously to prevent dehydration. Animals were carefully monitored post surgery and given free access to food and water. All animals undergoing surgery were allowed to recover for at least 4 days prior to the pilocarpine procedure.

**The pilocarpine model**. Briefly, rats were treated with pilocarpine (380 mg/kg i.p., Sigma) 20 min after scopolamine methyl nitrate (1 mg/kg i.p.). Diazepam (10 mg/kg i.p.) was administered ~2 h after the onset of status epilepticus and repeated as needed at 20-min intervals for a maximum of three doses. Rats recovering from status epilepticus were given a bolus of saline (0.9%) subcutaneously in the scruff and all animals had ready access to wet food and water. Cannulated animals were infused with 1 μl of either D-serine (100 μM, Sigma) or aCSF (vehicle) by means of a syringe pump (0.25 μl/min; KD Scientific) daily beginning with the last dose of diazepam for the subsequent 29 days. Following recovery from status epilepticus, rats were video-monitored (40 h/week) for frank spontaneous seizures.

**Brain fixation and slicing**. Rats were deeply anesthetized with urethane (1.5 mg/kg; i.p.) prior to intra-aortal fixation with 4% paraformaldehyde (PFA) in a 0.1 M phosphate buffer solution (PB; pH 7.4; 4 °C) following an initial flush with ice-cold saline (0.9%, 4 °C). Brains were removed and post-fixed overnight in PFA before being transferred to a 30% sucrose solution in PB until equilibration. They were then transferred to a mold, covered in O.C.T. solution (Tissue-Tek), and stored at −20 °C until sectioned. Semi-horizontal slices (50 μm thick) were cut on a cryostat and the sections (six series comprising of 12 sections per series) collected in a cryoprotectant solution consisting of 30% ethylene glycol and 25% glycerol in 50 mM PB. The cut sections were stored at −20 °C until processed or analyzed.

**Stereological analysis of MEA**. A one-in-six series of brain sections (50 μm) from a rat was ordered along a dorsal-ventral axis, washed in PB, and Nissl stained with thionin (1:100 of 0.25% stock solution) for 4 min under constant agitation. Sections were subsequently rinsed in PB for 2 min before being wet-mounted individually on glass slides and cover-slipped. The MEA in each lobe was demarcated on a microscope (Leica DM 5000B) using a 1.2× objective and the total number of neurons estimated from counts made with a 100× oil-immersion objective using the optical fractionator method. Stereo Investigator software (version 6.0, MBF, Colchester, VT) was used to specify sample sites (counting frame, 40 × 40 μm) on the Nissl-stained sections and an average of ~150 sites distributed across 12 sections were sampled in each hemisphere per rat. A guard zone of 10 μm upper and 20 μm lower was placed on each section in which no neurons were counted. Neurons were counted only if a nucleolus was visible. The average optical slice thickness measured was 48 μm over all the slices analyzed.

**Imaging**. *Nissl*: Cryo-protected brain slices fixed in PFA were Nissl stained with thionin (1:100 of 0.25% stock solution) at room temperature for 5 min, washed in PB, and affixed onto gelatin-coated glass slides through air drying. Slices were subsequently defatted for 1 h in 1:1 chloroform, 100% ethanol solution, then cleared in xylene (3 and 4-min rinses). The sections were then cover-slipped in Depex mounting medium. Images of the MEA were captured using Brightfield microscopy (Leica DM 5000B) at ×1.25, ×40, and ×100 magnification.

**Immunofluorescence**. Cryo-protected brain slices fixed in PFA were washed in PB (3 and 5-min rinses), main rinse solution (MRS: 0.1 M PB, 0.1 M glycine, 0.5% Triton X-100; 3 and 20-min rinses) before being exposed to a blocking solution (0.1 M PB, 0.5% Triton X-100, 2% goat serum, 2% bovine serum albumin) for a minimum of 1 h on a shaker. Slices were then exposed to the primary antibody in blocking solution overnight at room temperature. For double staining, we used the following combinations of primary antibodies: anti-NeuN Alexa-555, clone A60 (Millipore, mouse; MAB377A5; 1:500), and anti-GFAP Alexa-488 (Millipore, mouse; MAB3402X; 1:500) or anti-NeuN, clone 27-4 (Millipore, rabbit; MABN140; 1:500) and anti-CD11b/c (OX42) (Abcam, mouse; ab1211; 1:200). Slices were then washed in PB (3 and 5-min rinses), MRS (3 and 20-min rinses) before being exposed to the secondary antibodies in blocking solution. The following secondary antibodies were used for NeuN clone 27-4 and OX42, respectively: goat anti-rabbit Alexa-405 (Invitrogen, A-31556; 1:200) and goat anti-mouse Alexa-488 (Invitrogen; A-11001; 1:200). For the triple stains, we used anti-GFAP Alexa-488, anti-NeuN Alexa-555, and anti-CD11b/c (OX42) as our primary antibodies and goat anti-mouse biotin (Invitrogen, B-2763; 1:200) and streptavidin Alexa-594 (Invitrogen, S11227; 1:1000) as our secondary antibodies for OX42. To avoid cross-labeling of the secondary antibodies, as all primary antibodies were raised in mouse, slices were exposed to the OX42, biotin and streptavidin before the rest of the fluorophore-conjugated anti-NeuN and anti-GFAP primary antibodies were added to the mixture overnight. Slices were finally washed in MRS (6 and 10-min rinses) then mounted on slides using vectashield mounting media with or without DAPI (Vector Laboratories, CA). For serine racemase immunoreactivity, we used a mouse monoclonal anti-serine racemase antibody (GeneTex; GTX83567, 9E8; 1:200) to which brain slices were exposed overnight followed by goat anti-mouse biotin (Invitrogen, B-2763; 1:200) and streptavidin Alexa-594 (Invitrogen, S11227;

1:1000) secondary antibodies (same as for OX42). To avoid cross species labeling, as before, serine racemase, biotin, and streptavidin were immunoreacted first, followed by anti-NeuN and anti-GFAP primary antibodies, which were added overnight. A serine racemase primary-negative was also undertaken as control. Slices were finally washed in MRS (6 and 10-min rinses) then mounted on slides using vectashield mounting media with or without DAPI (Vector Laboratories, CA). Slices were imaged on a confocal laser-scanning microscope (Zeiss LSM 880) using an EC Plan-Neofluar 10×/0.30 WD = 5.2 M27 or Plan-Apochromat 63×/1.40 Oil DIC M27 objectives with appropriate excitation/emission filters. The 63× images were captured using a 35 μm Z-stack with images taken at 1.5 μm intervals. Validation of antibodies was provided by the vendor.

**Glial cell counting and Sholl analysis**. Astrocyte (GFAP) and microglia (OX42) densities in layers 2 and 3 of the MEA were determined from confocal-acquired images (10×) using the Fiji-ImageJ (version 1.52) cell counting tool. Neuronal (NeuN) staining was used to demarcate layer boundaries. GFAP-stained astrocytes were processed for morphometric changes in their arborization using Sholl analysis with the aid of the Simple Neurite Tracer (SNT) plugin for Fiji-ImageJ as described and validated in Tavares. et al.[19]. The Sholl analysis utilized high magnification images (63×) acquired on the confocal microscope and three to five clearly delineated cells in each brain slice were analyzed. The SNT plugin enabled tracing of astrocytic processes through individual images of the Z-stack to determine total number of radial crossings as a function of distance from origin (soma). Changes in the number of radial crossings were indicated by changes in color of the concentric Sholl circles drawn at 4 μm intervals. Cell volumes were also determined from the tracings with threshold set at 0.05 μm³. Note that astrocytic morphology can be complex with a preponderance of fine processes that can only be revealed through additional staining[31]. Hence, the morphometric measurements referred to in this manuscript pertain exclusively to astrocytes labeled with GFAP alone.

**Cell biology**. *Tissue collection*: Briefly, rats were deeply anesthetized with urethane (1.5 g/kg i.p.), decapitated, and horizontal slices (600 μm thick) were cut from the excised brains in ice-cold "cutting solution" containing the following (in mM): 230 sucrose, 10 D-glucose, 26 NaHCO₃, 2.5 KCl, 1.25 NaH₂PO₄, 10 MgSO₄, and 0.5 CaCl₂ (equilibrated with 95% O₂/5% CO₂). The MEA in each slice (six to eight slices per brain weighing between 60 and 80 mg) was carefully resected and collected in the cutting solution. Cerebellar samples were also collected for use as neuronal control tissue. Following removal of cutting solution, the tissue was washed in the cell wash buffer and exposed to the cell permeabilization buffer (250 μl; Mem-PER plus kit, Thermo Fisher) with a protease inhibitor cocktail (1X, Sigma) and MG132 (10 μM, Sigma). The tissue was subsequently frozen on dry ice and stored at −80 °C until further use. Immunoblotting: Frozen tissue suspended in the cell permeabilization buffer was thawed on ice and homogenized using a dounce homogenizer (Kontes) before being incubated at 4 °C for 10 min under constant rotation. The samples were subsequently fractionated using centrifugation (16,000 × g for 15 min at 4 °C). The supernatant, containing the cytoplasmic fraction, was mixed with 4X SDS buffer (in mM): 100 Tris (pH 6.8), 5 EDTA (pH 8), 4% SDS, 5% β-mercaptoethanol, 0.1 mg/ml bromophenol blue, boiled for 3 min at 95 °C, and loaded onto a 10% bis-acrylamide gel and run for 70 min at 180 V in 1X running buffer (in mM): 125 Tris, 200 glycine, and 0.1% SDS. The gel was subsequently transferred, using the wet transfer technique in 1X transfer buffer (in mM): 125 Tris, 200 glycine, 20% methanol, and 0.04% SDS to nitrocellulose membrane for 80 min (1.2 A, 4 °C). The nitrocellulose membrane was then stained with Ponceau S and blocked in 5% milk in Tris-buffered saline (TBS), Tween-20 (TBST) for 30 min. Antibodies were added at a dilution of 1:1000 in 0.5% milk dissolved in TBST and allowed to incubate overnight at room temperature under constant agitation. The following antibodies were used (1:1000): anti-NeuN (rabbit, Millipore, MABN140); anti-GFAP (rabbit, Abcam, ab206586); anti-IbaI (rabbit, Wako, 016-20001); and anti-GAPDH (rabbit, Sigma, G9545. 1:2000). Membranes were subsequently washed (3 and 5-min washes in TBST) and incubated with an appropriate secondary antibody [goat anti-rabbit (IRDye 800CW), at 1:15,000 dilution in TBST] for 1 h at room temperature under constant agitation. The membranes were subsequently rewashed in TBST (3 and 5-min washes) followed by TBS (2 and 5-min washes) before being imaged on an Odyssey infrared imaging system (LI-COR). Validation of antibodies was provided by the vendor.

**D-serine analysis**. Tissue was collected as described for immunoblotting. The MEA, LEA, and cerebellar samples were maintained in ice-cold cutting solution, weighed and transferred to 200 μl of HEPES-buffered aCSF (in mM): 126 NaCl, 10 D-glucose, 26 NaHCO₃, 3 KCl, 1.25 NaH₂PO₄, 2 MgSO₄, and 2 CaCl₂, 10 HEPES (pH 7.4), and incubated for 30 min at 32 °C. The supernatant was removed, centrifuged (12,000 × g for 10 min at 4 °C) and frozen at −80 °C until analyzed. The remaining tissue was homogenized in 300 μl of ice-cold aCSF with protease inhibitors (Sigma). The homogenate was centrifuged (12,000 × g for 10 min at 4 °C) and the supernatant removed and frozen at −80 °C until analyzed.

Briefly, stock solutions of 36 derivatized[23] primary amines were made in DI water including: L-Ser, D-Ser, L-threonine (L-Thr), D-Thr, L-asparagine (L-Asn), L-glutamine (L-Gln), D-Gln, L-Glu, D-Glu, L-Ala, D-Ala, L-histidine (L-His), L-Asp, D-Asp, L-tyrosine (L-Tyr), D-Tyr, GABA, L-valine (L-Val), D-Val,

L-methionine (L-Met), D-Met, L-isoleucine (L-Ile), L-leucine (L-Leu), D-Leu, L-phenylalanine (L-Phe), D-Phe, L-tryptophan (L-Trp), D-Trp, L-arginine (L-Arg), D-Arg, taurine (Tau), β-Ala, β-Phe, alpha-aminobutyric acid (α-ABA), β-homoserine (β-HSer), and Gly. From the stock solutions, the primary amines were further diluted in aCSF for calibration curves. β-Phe, α-ABA, and β-HSer were used as internal standards (IS). Sodium tetraborate buffer (15 mM, pH 9.0) was used to make up a sodium cyanide (NaCN, 40 mM) stock solution and to dilute 2,3-naphthalenedicarboxaldehyde (NDA, Thermo Fisher Scientific, Waltham, MA) to 5 mM in a 1:1 (v:v) mixture with acetonitrile. Samples and standards were derivatized in a 20:1:1:2 (v:v:v:v) ratio of sample: 40 mM NaCN: 24 µM IS: 5 mM NDA in a 0.2 mL polypropylene tube. The tube was vortexed for 30 s and then placed in the sample tray of the CE instrument to allow the labeling reaction to proceed for 30 min prior to injection. MEKC experiments were carried out on a Beckman Coulter PA800 with a laser-induced fluorescence module. Separation buffer consisted of boric acid (150 mM, pH 9.2), sodium dodecyl sulfate (45 mM) and sodium deoxycholate (35 mM). A fused-silica capillary (total length: 60 cm, inner diameter: 25 µm; Polymicro Technologies, Phoenix, AZ) was used for separation with an applied 29 kV separation voltage. A laser diode (100 mW, 450 nm; Lilly Electronics, Hubei, P.R. China) was used for detection. The laser power prior to the capillary was measured to be ~1 mW. Emission was filtered by 480 ± 20 nm bandpass and 457 nm notch filters prior to detection. Data were acquired at 4 Hz.

The initial daily rinse protocol consisted of 10-min 50 psi rinses each of 1 M NaOH, 0.1 M NaOH, and DI water. Between analyses, the capillary was rinsed with DI water for 10 min at 50 psi, held in DI water vials for 10 min, and then re-rinsed with 0.1 M NaOH followed by the running buffer, each for 5 min at 50 psi. Samples were injected hydrodynamically by applying 1 psi for 15 s to the sample tube, followed by dipping the ends of the capillary in DI water vials for 6 s to eliminate excess sample at the inlet of the capillary. The end of day rinse protocol consisted of 10 min 50 psi rinses of 0.1 M NaOH and DI water. The capillary was stored in DI water vials when not in use. Cartridge and sample storage temperatures were held at 25 °C for all runs.

Peak area and migration times were calculated using 32 Karat™ software (Beckman Coulter, Brea, CA). All peak areas were background subtracted using a blank electropherogram and then normalized to the peak area of a background-subtracted IS. The peak areas of L-Ser, D-Ser, L-Thr, D-Thr, L-Asn, L-Gln, D-Gln, L-Glu, and D-Glu were compared to β-HSer; the peak areas of L-Asp, D-Asp, L-His, Gly, L-Ala, D-Ala, L-Tyr, D-Tyr, L-Met, D-Met, β-Ala, L-Val, D-Val, and Tau were compared to α-ABA; the peak areas of GABA, L-Ile, L-Leu, D-Leu, L-Phe, D-Phe, L-Trp, D-Trp, L-Arg, and D-Arg were compared to β-Phe. Quantitation of these peak areas was performed using calibration curves obtained each day. Regression equations were calculated by linear least squares. Limits of detection were calculated by the concentration that was equivalent to three times the standard deviation of the background peak area divided by the slope of the corresponding calibration curve.

Assessment of D-serine levels in brain tissue was performed blind, wherein those processing the tissue for MEKC were unaware of whether the sample came from a control or epileptic animal.

**Statistical tests**. Unless otherwise noted, statistical significance was measured with the paired or two-sample equal variance (homoscedastic) Student's $t$ test using the two-tailed distribution. Error bars in the figures represent standard error of the mean (SEM). Statistical analysis was done using GraphPad Prism (version 7).

**Reporting summary**. Further information on research design is available in the Nature Research Reporting Summary linked to this article.

## Data availability
All data supporting the findings of this study are available within the article and its Supplementary Information. Source data are provided with this paper. All video recordings documenting seizure behavior can be made available upon request to the corresponding author. All other data, material, and reagents are available upon reasonable request to the corresponding author. Source data are provided with this paper.

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

## Acknowledgements

We wish to acknowledge with gratitude Dr. Frank Johnson's guidance with stereology — this work would not have been possible without his help and support. Our thanks are due to Dr. Saad Abbasi and Dr. Tiffany Jacobson for their efforts in getting this project started and Dr. Florian Duclot for initial guidance with stereotaxic surgery. We are most grateful to Mrs. Jenny Beesley for her help and guidance on this project. We are indebted to Ruth Didier for her expert guidance and assistance with confocal microscopy and Dr. Michael Blaber for his critical feedback on our manuscript. This work was supported in part by grants from the CRC and CoM at Florida State University, Epilepsy Foundation, and the National Institutes of Health (R01NS097802) to S.S.K and (R21DA044442) to M.G.R.

## Author contributions

S.S.K. and S.B. designed and analyzed all experiments outlined in this manuscript. S.B. contributed to stereotaxic surgeries, cell biology, stereology, and histology. T.S. contributed to confocal imaging and K.C. to stereology. R.A. helped with histology and cell counting. C.D'O. contributed to video monitoring along with K.C., K.E., E.O.O., and M.G.R. contributed to the assessment of D-serine levels in brain tissue. S.S.K., S.B., T.S., K.C., R.A., and C.D'O. all contributed to bringing up the pilocarpine model of TLE. T.S. and S.S.K. contributed to the electrophysiological assessment of the t-NMDAR function.

## Competing interests

The authors declare no competing interests.
