## [Peer Review File · Nature Communications]

Reviewers' comments:

Reviewer #1 (Remarks to the Author):

This paper examined mesiotemporal lobe epilepsy (MTLE) induced by the commonly used convulsant pilocarpine in an adult rat model. The authors hypothesized that a recently discovered triheteromeric NMDA receptor may be at the heart of neuronal cell loss in MTLE which presumably gives rise to epilepsy. The mechanistic explanation would be a differential effect of D-serine which is a coagonist on typical NMDA-R yet an antagonist of triheteromeric NMDA-R.

The authors present convincing data that

1. neuronal cell death occurs in the hippocampus following pilocarpine
2. that there is a delayed relative astrogliosis and microgliosis
3. that D-serine is lower in epileptic animals than in non-epileptic
4. that infusion of D-serine prevents neuronal death, astrogliosis and development of epilepsy as well as reduces seizure burden.

What is unclear is

1. which neurons are affected. Are these GABAergic, glutamatergic, do they express triheteromeric NMDAs.
2. What does the gliosis have to do with it. Is it a consequence (most likely) or cause? Do astrocytes or microglia provide D-serine and why might they fail after pilocarpine?
3. How is unilateral treatment affect the contralateral brain? The arguments provided are not strong.
4. Is the reactive astrogliosis and microgliosis present on both hemispheres and is it equally affected by unilateral D-serine treatment?

Taken together this is a very interesting paper that deserves to be shared with a larger readership. However the above additional mechanistic insights would elevate this paper from an observational study to a mechanistic study. Currently it shows D-serine supplementation preventing neuronal loss and epilepsy in a rat epilepsy model. If additional experiments can unequivocally demonstrate this to be due to D-serine insufficiency in reactive astrocytes as a consequence of seizures and mediated by selective protection of triheteromeric NMDA-Rs, this study would be more impactful.

Reviewer #2 (Remarks to the Author):

In the manuscript entitled "D-serine Mitigates Cell Loss Associated with Temporal Lobe Epilepsy", Bessley and collaborators investigate the possible implication of D-serine in the loss of layer 3 neurons in the medial entorhinal area, in the context of temporal lobe epilepsy, and its potentiality as a treatment.

The goal of the study is interesting. However, several experiments and statistics are missing to support authors conclusions.

To begin, it is important to notice that the article is not divided into distinct parts, making the reading complicated. In addition, while the introduction is almost absent, the first 6 lines of the second paragraph, page 2, are identical to the second paragraph of the paper published by the same group in 2019 in J. Neurophysiol. Authors have to modify the text.

Major points:

1- In the first paragraph of the "potential" introduction part, the following sentence is not clear: "The recent discovery of a novel class of D-serine-sensitive N-methyl-D-aspartate receptors (NMDARs)...". Indeed, it is known since several years that D-serine is a co-agonist of NMDARs. Please clarify this sentence.

2- In the first part of the results section, authors should indicate in the text, when needed, the Figure 1 and its associated panel.

- Importantly, authors have to cite the papers they are referring to in the following sentence: "The non-cannulated cohort of animals include those utilized in past studies of TLE in the laboratory" (legend Figure 1).

- In figure 1, the panel b is complicated to understand quickly. Is it possible to make it simpler?

3- In the second part, on the impact of D-serine treatment on neuronal loss, authors claim, page 4, lines 15-16, "Non-status animals receiving either D-serine or aCSF were normal in every aspect and could be distinguished from post-status animals based on the pattern of weight gain following initial insult (Fig. 2b)." However, no statistics support this conclusion. Please indicate it.

- In Figure 2d, if $p < 0.002$, why no stars are indicated on the bar graph? Also, for a better understanding, the supplementary figure 2 should appear in Figure 2.

- The panel 2f is not convincing. The loss of neurons in post-status aCSF is not visible.

- Figure 2h: authors indicate in the figure legend that "**** $p < 0.001$, **** $p < 0.0001$ ", however on the figure they have not indicated any star. In addition, if the numbers in the bars indicate the number of animals, the n of 1, 2 and 3 are too small.

- For a better understanding, authors should indicate p and n in the text. In addition, it would be better if authors indicate the "sem" with 2 digits after the decimal point.

- Authors claim page 5, last sentence, that D-serine treatment rescue neurons, however is it really a rescue? In addition, it should be interesting to see statistical results on the bar graph between "non-status" and "post-status".

4- In the next part on the investigation of the impact of D-serine on glial cells, authors are talking about infiltration of astrocytes, but again this is not convincing. Are they really infiltrating? From where? - The use of GFAP to count the number of astrocytes is probably not a good option. Indeed, it is complicated to localize the cell body with this labeling, as GFAP is an intermediate filament protein whose expression changes when astrocytes are reactive. In addition, only 10% of the cell is labeled with GFAP (Bushong et al., 2002). Therefore, authors should use in addition to GFAP labeling, which confirms that the cell is an astrocyte, another protein like S100beta or ALDH1L1 (alone).

- Apparently, using this GFAP labeling, astrocytes look still reactive with D-serine treatment, and the number of astrocytes looks still important in specific locations. Why? Authors should investigate and discuss this.

In Figure 3, panels b and c, authors should indicate the number of animals and sections in all bars. In addition, to what correspond the blue in f and g?

In the same figure, panel c, authors indicate the number of microglia while no images are presented in this figure, only in the next one, Figure 4. Authors should indicate this bar graph in Figure 4.

- In Figure 4 the labeling of microglia in non-epileptics is not visible, why? Microglial cells should be visible with thin processes. In addition, microglial cells look over activated in epileptic as well as in non-epileptic in D-serine-treated mice. Authors should discuss this.

- Why authors claim that microglia "do not bring about epilepsy?" This conclusion is based on which data?

5- Part on D-serine levels and summary of neuro-glia changes in TLE.

- In Figure 5, why traces are shifted on the right?

- In the text authors indicate that there is a significant decrease of D-serine. This is not visible in Figure 5, panel b. Again, in the legend, Figure 5, authors indicate "* $p < 0.05$, *** $p < 0.001$, Student's t-test", while no stars are present in bar graphs. Authors should use another statistical test, in regards to the small number of samples.

- What does the D-serine treatment on the amount of D-serine in the tissue?

6- Additional questions:

- Which cell is capable to synthesize D-serine in the MEA? Authors should perform immunohistochemical experiments to localize serine racemase.

- In addition, do astrocytes express NMDARs in MEA?

- What are the clues leading the authors to propose a switch of neuroprotective role? They should discuss this more.

Minor points:

- The bibliographic references appear in the abstract. Authors, should modify this.

- Figure 2g: "micro" for micrometer is missing.

- Title Figure 4: "microglia" and not "micorglia".

RESPONSE TO REVIEWERS CONCERNS & MODIFICATIONS MADE TO THE MANUSCRIPT

Through this revision, we have addressed most, if not all, of the issues raised by the reviewers. Several new experiments have been undertaken to directly answer the questions raised in order to bolster the conclusions. A significant amount of resources have been dedicated for this purposes and we hope the revisions affected will be viewed in the positive light.

We are indeed grateful to both the reviewers and the editors for their insightful comments and suggestions regarding improving the scope and appeal of our manuscript- thank you!

New experiments undertaken include:

- 1) Assessment of the source of D-serine (neurons and/or astrocytes) through serine racemase immunohistochemistry. We were clearly surprised by the fidelity of these experiments that show unequivocally that both neurons and astrocytes in the MEA have the ability to make D-serine. A new figure has now been added to the manuscript to showcase these important studies.
- 2) Quantification of astrocytes in both left and right hemispheres to show that the bilateral effects of D-serine, applicable to neurons, are recapitulated with astrocytes. A new supplementary figure quantifying these data has now been added to the manuscript.
- 3) Despite the fact that loss of GABAergic neurons in the MEA has been confirmed and reported in literature (kindly see my paper in Journal of Neuroscience), reviewer #1 asked for this specifically. We therefore undertook new experiments to ID the state of GABA neurons in the MEA with the best antibody against GAD65 available commercially. These experiments unfortunately did not yield any new data due to poor immunoreactivity/staining. We know from decades of experience that GABAergic neurons, for reasons unknown, stain very poorly with GAD antibodies and this was also the case here. We have therefore relied on data from our previous work using in situ hybridization methods, to address the issue of GABA neurons. Additionally, this data does not alter any of the conclusions made in the manuscript.
- 4) We undertook new experiments to ascertain whether other commercially available antibodies that detected astrocytes, bettered the fidelity of the morphometric data obtained through GFAP (gold standard), as reported in our manuscript originally and as requested by reviewer #2. We tried the better of the two antibodies suggested (ALDH1L1) and were thoroughly disappointed. Although, we were able to confirm co-localization with GFAP, the staining pattern with the new antibody precluded any quantification or morphometric assessments. Again, these data do not alter any conclusions drawn in the original manuscript.

The revised manuscript has now been reformatted as per Nature Communication guidelines and the figures now address the statistical data that quite unfortunately got left out during conversion to the PDF format during our original submission. Addressed below, are the specific concerns raised by the reviewers and the modifications we have made to our manuscript in light of these suggestions.

Reviewers' comments:

Reviewer #1 (Remarks to the Author):

This paper examined mesiotemporal lobe epilepsy (MTLE) induced by the commonly used convulsant pilocarpine in an adult rat model. The authors hypothesized that a recently discovered triheteromeric NMDA receptor may be at the heart of neuronal cell loss in MTLE which presumably gives rise to epilepsy. The mechanistic explanation would be a differential effect of D-serine which is a coagonist on typical NMDA-R yet an antagonist of triheteromeric NMDA-R. The authors present convincing data that

1. neuronal cell death occurs in the hippocampus following pilocarpine
2. that there is a delayed relative astrogliosis and microgliosis
3. that D-serine is lower in epileptic animals than in non-epileptic
4. that infusion of D-serine prevents neuronal death, astrogliosis and development of epilepsy as well as reduces seizure burden.

>> We are grateful for this positive feedback.

What is unclear is

1. which neurons are affected. Are these GABAergic, glutamatergic, do they express triheteromeric NMDAs.

>> Both excitatory and inhibitory neurons in the MEA are lost in TLE. The loss, specifically of GABAergic neurons, in the MEA has been confirmed and reported in the literature. Kindly see:

Kumar SS, Buckmaster PS (2006) Hyperexcitability, interneurons, and loss of GABAergic synapses in entorhinal cortex in a model of temporal lobe epilepsy. *J. Neurosci.* 26:4613-4623.

Although both neuron types are affected, the mechanisms underlying their loss are deemed different. Through this work, we showcase one avenue through which a majority of excitatory neurons in MEA are lost. However, the state of NMDARs in GABAergic neurons is not known (these are difficult neurons to record from). Our hypothesis is that GABAergic neurons in the MEA perish because of enhanced of presynaptic inhibition from interneurons in the adjoining pre- and parasubiculum that are rendered hyperexcitable in TLE.

We have now mentioned about the status of GABAergic neurons in the revised manuscript referencing our earlier work (Page 6, Paragraph 2). In addition, we also undertook fresh

experiments to ID the state of GABA neurons in the MEA with the best commercially available antibody against GAD65. These experiments unfortunately did not yield any new data due to poor immunoreactivity/staining. GABAergic neurons, for reasons unknown, stain very poorly with GAD antibodies and this was also the case here. We have therefore relied on data from our previous work using in situ hybridization methods, to address the issue of GABA neurons.

Note diffuse background staining with the GAD antibody (green) but lack of co-localization with NeuN (neuronal marker, red) either in low mag or high mag.

2. What does the gliosis have to do with it. Is it a consequence (most likely) or cause? Do astrocytes or microglia provide D-serine and why might they fail after pilocarpine?

>> We believe that neurodegeneration and gliosis work hand in glove to bring about the pathology in the MEA, and in the long run, feed into each other in a vicious cycle, disrupting D-serine availability and homeostasis to bring about neuroinflammation. Thus, the initial hyperexcitability induced loss of neurons might serve as a trigger for a cascade of events leading up to this state. This is now stated explicitly in the manuscript (Page 11, Paragraph 2).

To specifically address the question of whether gliosis is the consequence or the cause of neurodegeneration, we undertook western blot analysis of neuronal astrocyte and microglia markers immediately after status epilepticus. Our results show that neuronal loss in MEA precedes neuroinflammation and that it is manifest as early as one day following initial insult in post-status animals compared with the non-status controls (upper panel, Fig. 4e). Unlike neurons, astrocytes and microglia were not manifest in the MEA until post-insult day five at which point there was a marked increase in their relative abundance. This point is now reemphasized by the addition of a separate subsection in the results portion of the manuscript (Page 8, Paragraph 2).

Microglia can potentially be a source of D-serine, but in our case, these cells are only manifest in the epileptic MEA, thereby precluding the possibility they contribute in any major way to epileptogenesis (Page 11, Paragraph 2). Additionally, a recent study (now cited) also showed that two microglia specific markers (Iba1 and IB4) did not co-localize with serine racemase. To answer the question whether astrocytes or neurons provide D-serine and why they might fail after pilocarpine, we undertook new experiments to assay for serine-racemase, the enzyme responsible for the synthesis of D-serine from L-serine (kindly see new Figure 5). Our data clearly show that both neurons and astrocytes in the MEA have the ability to make D-serine. Why they might fail after pilocarpine is hypothesized in the neuronal and glial perspectives in the discussion (Page 10, Paragraph 2).

3. How is unilateral treatment affect the contralateral brain? The arguments provided are not strong.

>> We noticed that interventions in one hemisphere always affected the other- a consistent observation! We do not know how this lateralization comes about other than through the ventricles and/or cross hemispheric connectivity of neuron between the left and right MEAs. This has now been mentioned in the manuscript (Page 5, Paragraph 2). Further investigation of this process requires experimentation that is well beyond the scope of the current manuscript. Additionally, this information does not change any of our observations or conclusions.

4. Is the reactive astrogliosis and microgliosis present on both hemispheres and is it equally affected by unilateral D-serine treatment?

>> We have now quantified astrocytes in both the left and right hemispheres in control and epileptic animals to show that the bilateral effects of D-serine, applicable to neurons, are recapitulated with astrocytes. A new supplementary figure quantifying these data has now been added to the manuscript. Unfortunately, we were not able to accurately quantify and compare microglia across hemispheres given their size and overwhelming density. Given the trends we have seen with neurons and astrocytes, it would be highly unlikely if the microgliosis observed in the cannulated right hemisphere did not also carry over to the contralateral side. This does not alter any of the conclusions drawn in our manuscript.

Taken together this is a very interesting paper that deserves to be shared with a larger readership. However the above additional mechanistic insights would elevate this paper from an observational study to a mechanistic study. Currently it shows D-serine supplementation preventing neuronal loss and epilepsy in a rat epilepsy model. If additional experiments can unequivocally demonstrate this to be due to D-serine insufficiency in reactive astrocytes as a consequence of seizures and mediated by selective protection of triheteromeric NMDA-Rs, this study would be more impactful.

>> In addition to directly demonstrating depletion of D-serine in epileptic tissue, we have now undertaken new experiments to locate its source i.e. neurons v astrocytes- an important advancement. The loss of neurons and the change in reactive status of astrocytes is consistent

with the model proposed and highlights the importance of supplementing D-serine as a therapeutic option for curtailing TLE.

Reviewer #2 (Remarks to the Author):

In the manuscript entitled “D-serine Mitigates Cell Loss Associated with Temporal Lobe Epilepsy”, Bessley and collaborators investigate the possible implication of D-serine in the loss of layer 3 neurons in the medial entorhinal area, in the context of temporal lobe epilepsy, and its potentiality as a treatment. The goal of the study is interesting. However, several experiments and statistics are missing to support authors conclusions.

To begin, it is important to notice that the article is not divided into distinct parts, making the reading complicated. In addition, while the introduction is almost absent, the first 6 lines of the second paragraph, page 2, are identical to the second paragraph of the paper published by the same group in 2019 in J. Neurophysiol. Authors have to modify the text.

>> We thank the reviewer for his positive comments. We are deeply apologetic though for the lapse in the statistical information in the figures and certain special symbols not being faithfully carried over during the conversion process to PDF. By the time this was caught and brought to the attention of the editors of the journal, it was too late. The manuscript, originally submitted to Nature, has now been reformatted as per Nature Communications guidelines. We deem it important to clearly specify how t-NMDARs are classified and have taken no chances to deviate from what has already been published. However, in the interest of not sounding repetitive, the text has now been reworded slightly, as suggested.

1- In the first paragraph of the “potential” introduction part, the following sentence is not clear: “The recent discovery of a novel class of D-serine-sensitive N-methyl-D-aspartate receptors (NMDARs)...”. Indeed, it is known since several years that D-serine is a co-agonist of NMDARs. Please clarify this sentence.

>> We agree with reviewer (thank you!). This has now been clarified in the manuscript (page,2 paragraph,1).

2- In the first part of the results section, authors should indicate in the text, when needed, the Figure 1 and its associated panel.

>> Figure 1 and its panels have been referenced in text of the manuscript.

- Importantly, authors have to cite the papers they are referring to in the following sentence: “The non-cannulated cohort of animals include those utilized in past studies of TLE in the laboratory” (Legend Figure 1).

>> References supporting this have now been cited.

- In figure 1, the panel b is complicated to understand quickly. Is it possible to make it simpler?

>> We have now simplified figure 1 by moving the information pertaining animals used for the behavioral studies to supplementary information.

3- In the second part, on the impact of D-serine treatment on neuronal loss, authors claim, page 4, lines 15-16, "Non-status animals receiving either D-serine or aCSF were normal in every aspect and could be distinguished from post-status animals based on the pattern of weight gain following initial insult (Fig. 2b)." However, no statistics support this conclusion. Please indicate it.

- In Figure 2d, if $p < 0.002$, why no stars are indicated on the bar graph? Also, for a better understanding, the supplementary figure 2 should appear in Figure 2.

>> We have now added statistics to Fig 2b to show the difference between control and post-status animals. We have also added the statistics to Fig 2d. Supplementary Fig 2 has now been incorporated into Figure 2 proper.

- The panel 2f is not convincing. The loss of neurons in post-status aCSF is not visible.

>> We believe the loss of neurons is quite obvious in the magnified images. Kindly zoom into the low mag image to fully appreciate the loss of neurons. Neurons are well demarcated from debris, which is typical of Nissl-stained sections, in the control images.

- Figure 2h: authors indicate in the figure legend that "**** $p < 0.001$, **** $p < 0.0001$ ", however on the figure they have not indicated any star. In addition, if the numbers in the bars indicate the number of animals, the n of 1, 2 and 3 are too small.

>> The statistics are now designated with proper stars. The numbers in the bar plot on the right of panel h are to show the behavioral outcomes of the animals assayed (for full disclosure) and are not meant for statistical comparison. The total number of animals used in this part of the study are indicated in the bar plot to the left of the panel.

- For a better understanding, authors should indicate p and n in the text. In addition, it would be better if authors indicate the "sem" with 2 digits after the decimal point.- Authors claim page 5, last sentence, that D-serine treatment rescue neurons, however is it really a rescue? In addition, it should be interesting to see statistical results on the bar graph between "non-status" and "post-status".

>> The p and n values are either indicated in the text of the manuscript or figures associated with the text in the manuscript in a way that does not clutter the text and the highlighting of the main points. For neuron counts the SEMs have been rounded up to a single cell. We are of the opinion that the word 'rescue' is appropriate here as the neurons would have otherwise perished without D-serine. The confusion between the 'non-status' and 'post-status' counts arose because of the missing stars which are now appended to the figures.

4- In the next part on the investigation of the impact of D-serine on glial cells, authors are talking

about infiltration of astrocytes, but again this is not convincing. Are they really infiltrating? From where? - The use of GFAP to count the number of astrocytes is probably not a good option. Indeed, it is complicated to localize the cell body with this labeling, as GFAP is an intermediate filament protein whose expression changes when astrocytes are reactive. In addition, only 10% of the cell is labeled with GFAP (Bushong et al., 2002). Therefore, authors should use in addition to GFAP labeling, which confirms that the cell is an astrocyte, another protein like S100beta or ALDH1L1 (alone).

>> Our data clearly shows that there is an increase in the number of astrocytes in the MEA under epileptic conditions. We do not know whether these are infiltrating or proliferating astrocytes (or microglia), the assessment of which is beyond the scope of the current work (both possibilities are now acknowledged in the manuscript and/or just alluded to as astrocytic density). GFAP is the most widely used marker for astrocytes in the literature, both for counting and morphometric assessments. In light of what has been reported in the literature and the reviewer's comments, we undertook new experiments with the recommended Aldh1L1 antibody (clone N103/39, Millipore; MABN495). We are sorry to report that this antibody yielded data that was nowhere close to the fidelity of GFAP. We did, however, note co-localization with GFAP, but morphometric analysis proved impossible.

Apparently, using this GFAP labeling, astrocytes look still reactive with D-serine treatment, and the number of astrocytes looks still important in specific locations. Why? Authors should investigate and discuss this. In Figure 3, panels b and c, authors should indicate the number of animals and sections in all bars. In addition, to what correspond the blue in f and g? In the same figure, panel c, authors indicate the number of microglia while no images are presented in this figure, only in the next one, Figure 4. Authors should indicate this bar graph in Figure 4.

>> Astrocytes are still in their reactive state even with D-serine treatment; however, the number of astrocytes in the MEA-layer 3 is significantly reduced compared with the aCSF treated animals. Astrocytes are known to exist in multiple reactive states (Liddelow SA, Barres BA, 2017) and this has been referenced in the manuscript (page 8, paragraph 1). Astrocytes in the D-serine treated MEA still appear to be reactive and restricted to the medial part of the MEA. This has been discussed in the manuscript (page 7, paragraph 1). The number of animals and sections used in generating the bar plots are clearly indicated in the figure. The blue in Fig 3f corresponds to D-serine post-status non-epileptic whereas, the blue in Fig 3g corresponds to all D-serine post-status animals. The microglia figure (Fig 4a) is now referenced with the bar plot in Figure 3 to reconcile this ambiguity in the interest of conserving space.

- In Figure 4 the labeling of microglia in non-epileptics is not visible, why? Microglial cells should be visible with thin processes. In addition, microglial cells look over activated in epileptic as well as in non-epileptic in D-serine-treated mice. Authors should discuss this.
- Why authors claim that microglia “do not bring about epilepsy?” This conclusion is based on which data?

>> We can only report what we can see with our immunohistochemistry and antibodies and not speculate on what we cannot see. Microglia are seen in tissue from epileptic animals but not from controls, using the same staining protocol. Furthermore, there is no way to distinguish ‘over-activated’ microglia from “regular” microglial and there is not much precedence for this in the literature in the context of cell loss in the MEA. We believe that all the microglia seen in the epileptic animals are of the over-activated kind. Additionally, we do not wish to dwell on the over-activatedness of microglia in the context of what we are trying to report in this manuscript. Microglia are most conspicuous after the animal has become epileptic (verified through western blot analysis as well as through imaging) suggesting that they are not the cause for epilepsy per se (page 8, paragraph 3).

5- Part on D-serine levels and summary of neuro-glia changes in TLE.

- In Figure 5, why traces are shifted on the right?
- In the text authors indicate that there is a significant decrease of D-serine. This is not visible in Figure 5, panel b. Again, in the legend, Figure 5, authors indicate “* $p < 0.05$, *** $p < 0.001$, Student’s t-test”, while no stars are present in bar graphs. Authors should use another statistical test, in regards to the small number of samples.
- What does the D-serine treatment on the amount of D-serine in the tissue?

>> The traces are time-shifted to the right to highlight amplitude differences and this is clearly stated in the figure legend (now Fig 6a). Statistical differences are now highlighted in the figures with appropriate statistical tests and sample sizes of n = 8 animals (control) and n = 6 animals (epileptic). The bars atop peaks facilitate the viewing of these amplitude differences. D-serine measurements were to assay ambient levels in control and epileptic MEA. See supplementary figure 4 for the effects of added D-serine on amplitude.

6- Additional questions:

- Which cell is capable to synthesize D-serine in the MEA? Authors should perform immunohistochemical experiments to localize serine racemase. - In addition, do astrocytes express NMDARs in MEA? - What are the clues leading the authors to propose a switch of neuroprotective role? They should discuss this more.

>> We thank the reviewer for the excellent suggestion. We too believe that knowing what the potential sources of D-serine are in the MEA significantly strengthens the impact of our work. Consequently, we undertook new experiments to answer this question using serine racemase, the enzyme that converts L to D- serine. We are happy to report that both astrocytes and neurons express serine racemase and could potentially be independent sources of D-serine in the MEA. A new Fig 5 has now been added to the manuscript and this information dovetails nicely with the hypothesis relating to the switch in the neuro-protective role of astrocytes.

Minor points:

- The bibliographic references appear in the abstract. Authors, should modify this.
- Figure 2g: "micro" for micrometer is missing.

>> Now corrected-thank you!

- Title Figure 4: "microglia" and not "micorglia".

>> Now corrected-thank you!

REVIEWER COMMENTS

Reviewer #1 (Remarks to the Author):

The authors have addressed my concerns in this revised manuscript. The work provides a plausible explanation for the excitotoxic neuronal loss being selective for certain subpopulations of neurons in the hippocampus and surrounding brain based on the action of D-serine. The work has significant translational potential given the finding that D-serine acts as enhancer of d-NMDA versus blocker at t-NMDA receptors.

Reviewer #2 (Remarks to the Author):

First of all, I would like to thank Dr Kumar and his collaborators for the additional work they have performed since the last submission. The paper is well improved, but I still have comments.

Four major points:

1) From the beginning of the abstract authors claim that both glia and neurons conspire to give rise to the pathology.

For sure the state of glial cells has changed. However, authors didn't really show that they conspire.

2) The second conclusion authors made is regarding the antagonism action of D-serine on t-NMDARs. First of all, authors should show that t-NMDAR activity is higher and d-NMDAR activity is smaller after an epileptogenic status. If this is true this should be rescued by D-serine treatment. Authors have to perform additional electrophysiological experiments to make this strong conclusion.

3) The third conclusion is regarding the attenuation of proliferation and/or infiltration of microglia. Page 8, lines 205-206 authors wrote: "Microglia, essentially absent in the non-status controls, were strikingly manifest in epileptic animals (top panels, Fig. 4a)".

The effect they report on the number of microglia could also be due to an effect on the expression level of microglial proteins they looked at. Authors should perform additional experiments or at least discuss it.

4) Finally, how authors can determine a temporal sequence using only Western-Blots? They should find a second way to show it and discuss the limits of the methods they have used.

Minor points:

5) In the panel a of the figure 1, I understand that 14 animals were implanted. Does that mean that 8 animals were injected with aCSF and 8 with D-serine? In the excel file, 17 animals are presented. How many animals were not implanted, implanted and how many had an osmotic pump?

I have the same comment for the Figure 2a.

6) In the legend of Figure 2h, could you please indicate to what correspond the numbers in bars.

7) Page 5, lines 130-131: "Post-status animals treated with D-serine resembled controls with only a moderate loss of neurons (Fig 2f)."

Did authors perform statistics to conclude this? It is surprising to make this conclusion with $n = 2$.

8) Page 6, lines 164-165 authors wrote: "This pattern was strikingly altered in the epileptic

animals and it appeared as though the entire layer 3 neuronal population was replaced by astrocytes (top panels, Fig. 3a)."

Could authors represent the number of neurons (Figure 2), astrocytes and microglia (Figure 3) in the same way please?

9) Page 8, lines 194-196: "We found a significant increase in astrocytic volume in MEA of epileptic animals compared with the non-epileptic controls (control: 35 ± 6 versus epileptic: $91 \pm 11 \mu\text{m}^3$, $p < 0.001$, Fig. 3e), despite a similar degree of arborization¹⁹ (Fig. 3f)."

Using GFAP labeling, authors cannot measure the astrocytic volume as GFAP only represents 10 % of the volume of the astrocyte (Bushong et al., 2002). It is a GFAP volume.

RESPONSE TO REVIEWERS' CONCERNS & MODIFICATIONS MADE TO THE MANUSCRIPT (Round 2)

Through this second-round revision, we are hoping to address the remaining issues / concerns of one of two reviewers of our manuscript. We are eager to bring closure to this review process especially in light of several new experiments undertaken and completed in round-1 to strengthen our findings. This revision, in our opinion does not require any additional experiments, and we have tried to be as responsive to the reviewers' suggestions as is possible. We are grateful to the reviewers and editors for their efforts in strengthening our manuscript. We look forward to a quick final decision- thank you!

REVIEWER COMMENTS

Reviewer #1 (Remarks to the Author):

The authors have addressed my concerns in this revised manuscript. The work provides a plausible explanation for the excitotoxic neuronal loss being selective for certain subpopulations of neurons in the hippocampus and surrounding brain based on the action of D-serine. The work has significant translational potential given the finding that D-serine acts as enhancer of d-NMDA versus blocker at t-NMDA receptors.

>> Thank you!

Reviewer #2 (Remarks to the Author):

First of all, I would like to thank Dr Kumar and his collaborators for the additional work they have performed since the last submission. The paper is well improved, but I still have comments.

>> Thank you for all your efforts in helping us improve our manuscript! We are grateful.

Four major points:

1) From the beginning of the abstract authors claim that both glia and neurons conspire to give rise to the pathology.

For sure the state of glial cells has changed. However, authors didn't really show that they

conspire.

>> We have now dropped the word “conspire” altogether and chosen to use “together” instead.

2) The second conclusion authors made is regarding the antagonism action of D-serine on t-NMDARs. First of all, authors should show that t-NMDAR activity is higher and d-NMDAR activity is smaller after an epileptogenic status. If this is true this should be rescued by D-serine treatment. Authors have to perform additional electrophysiological experiments to make this strong conclusion.

>> Electrophysiological evidence suggests that ~80% of neurons in the MEA express t-NMDARs (assayed through multiple approaches including molecular biology and characterization of current-voltage relationship using electrophysiology) and we are not quite sure how the conclusion that epileptogenic activity can alter this phenotype was reached. These findings have been published (Kindly see Beesley S et al. (2019) J Neurophysiol. 121: 238-254; Pilli J, Kumar SS (2012) Neuroscience 222:75-88).

3) The third conclusion is regarding the attenuation of proliferation and/or infiltration of microglia.

Page 8, lines 205-206 authors wrote: “Microglia, essentially absent in the non-status controls, were strikingly manifest in epileptic animals (top panels, Fig. 4a)”.

The effect they report on the number of microglia could also be due to an effect on the expression level of microglial proteins they looked at. Authors should perform additional experiments or at least discuss it.

>> We have both a positive and negative result with the same antibody under two different conditions validating the efficacy of the antibody in identifying microglia. Secondly, there is no reason to assume that a negative result with the antibody equates to low expression level of microglial proteins given that CD11b/OX42 is a common integrin protein that is expressed on the surface of many leukocytes including macrophages and microglia. This protein has been assayed in other studies as an identifier of microglia and found data that is similar to what we have reported (Drexel M, Preidt AP, Sperk G. Neuropharmacology 63, 806-817 (2012). Thirdly, as indicated in the previous round of reviews, we can only report what we can see with our immunohistochemistry and antibodies and not speculate on what we cannot see even though we did try alternate antibodies (Iba1, from Millipore and WAKO) for visualization. Finally, even if additional microglia in the control are revealed hypothetically, this does not alter any of our conclusions.

4) Finally, how authors can determine a temporal sequence using only Western-Blots? They should find a second way to show it and discuss the limits of the methods they have used.

>> The western blot analysis reported in the paper is key in the determination of the temporal sequence of events and we do not know what's wrong with this data. To the best of our knowledge, this Herculean undertaking has been attempted and/or being reported for the first time in the literature making this manuscript worthy of consideration in Nature Communications.

We know of no other means other than cell counting (which has already been reported) to demonstrate this more directly or more unambiguously. Additionally, this issue was addressed to some extent in response to reviewer 1 (comment #2).

Minor points:

5) In the panel a of the figure 1, I understand that 14 animals were implanted. Does that mean that 8 animals were injected with aCSF and 8 with D-serine? In the excel file, 17 animals are presented. How many animals were not implanted, implanted and how many had an osmotic pump? I have the same comment for the Figure 2a.

>> Kindly refer to Supplementary Figure 2 (black boxes) for animal usage in behavioral experiments. Figure 2a refers to animals used for histological studies. Out of 87 animals, 39 were infused with aCSF through cannula, 30 with D-serine through cannula and 18 with D-serine using osmotic pumps. The tab in the excel sheet for Fig. 2a refers to the pie chart for that figure and not the number of animals used.

6) In the legend of Figure 2h, could you please indicate to what correspond the numbers in bars.

>> The numbers in the bars indicate the total number of animals (n) used. This is now stated explicitly in the figure legend (Page 31, Paragraph 1).

7) Page 5, lines 130-131: "Post-status animals treated with D-serine resembled controls with only a moderate loss of neurons (Fig 2f)."

Did authors perform statistics to conclude this? It is surprising to make this conclusion with n = 2.

>> We are just describing our initial observations here (Figures 2f, 2g). The statistics are in 2h. The numbers in the bar plot on the right of panel h are to show the behavioral outcomes of the animals assayed (for full disclosure) and are not meant for statistical comparison. The total number of animals used in this part of the study are indicated in the bar plot to the left of the panel. This has now been stated explicitly in the figure legend (Page 31, Paragraph 1).

8) Page 6, lines 164-165 authors wrote: "This pattern was strikingly altered in the epileptic animals and it appeared as though the entire layer 3 neuronal population was replaced by astrocytes (top panels, Fig. 3a)."

Could authors represent the number of neurons (Figure 2), astrocytes and microglia (Figure 3) in the same way please?

>> The statement made, although descriptive to draw the attention of the reader, is factually correct. The quantitation is presented in the next paragraph. Is this in reference to the color difference between the histograms or to the left-right hemispheric differences, which are now shown in supplementary Figure 3?

9) Page 8, lines 194-196: "We found a significant increase in astrocytic volume in MEA of epileptic animals compared with the non-epileptic controls (control: 35 ± 6 versus epileptic: $91 \pm$

11 μm^3 , $p < 0.001$, Fig. 3e), despite a similar degree of arborization¹⁹ (Fig. 3f).”
Using GFAP labeling, authors cannot measure the astrocytic volume as GFAP only represents 10 % of the volume of the astrocyte (Bushong et al., 2002). It is a GFAP volume.

>> The Bushong 2002 work compares fluorescent dye-filled and GFAP labeled astrocytes revealing that GFAP staining only demarcates ~15% of the cell. Finer processes can only be revealed through filling individual astrocytes with a battery of fluorescent dyes (including Alexa 568, 488 or Lucifer yellow). This is neither practical for this study nor the intent of this study. Furthermore, there is no clear data on how changes in the reactive status of astrocytes impacts the fine processes-it may or may not-and hence this issue is not central to our study when reactive status can be gauged with GFAP alone, as has been the precedence in the literature. Additionally, as per prior recommendation, we have tried to use an alternate antibody (Alzh1L1) to reveal these fine processes to no avail. This issue has been addressed in round one.

However, in deference to the reviewer's concern, we have now explicitly clarified that morphometric analysis of astrocytes in our manuscript is based on the structure revealed by GFAP staining alone and referenced the Bushong study (Page 19, Paragraph 1).

REVIEWER COMMENTS

Reviewer #2 (Remarks to the Author):

First of all, I would like to thank Dr Kumar and his collaborators to have taken into account my previous comments.

However, I still have a problem with the second major point I mentioned last time. It was the following:

2) The second conclusion authors made is regarding the antagonism action of D-serine on t-NMDARs. First of all, authors should show that t-NMDAR activity is higher and d-NMDAR activity is smaller after an epileptogenic status. If this is true this should be rescued by D-serine treatment. Authors have to perform additional electrophysiological experiments to make this strong conclusion.

Authors answer:

"Electrophysiological evidence suggests that ~80% of neurons in the MEA express t-NMDARs (assayed through multiple approaches including molecular biology and characterization of current-voltage relationship using electrophysiology) and we are not quite sure how the conclusion that epileptogenic activity can alter this phenotype was reached. These findings have been published (Kindly see Beesley S et al. (2019) J Neurophysiol. 121: 238-254; Pilli J, Kumar SS (2012) Neuroscience 222:75-88)."

It is clear that D-serine level is decreased in epileptic condition and that D-serine administration has a positive effect. However, authors didn't show that this positive D-serine effect is going through antagonism of t-NMDARs. I understand that it can be long to perform electrophysiological experiments. But, if this is true (antagonism of tNMDARs), D-serine effect should be reproduced by D-AP5 application. Indeed, authors mentioned in the introduction: "These receptors are blocked by the pan-NMDAR antagonist D-(-)-2-Amino-5-phosphonopentanoic acid (D-AP5) and by D-serine, a potential gliotransmitter and a co-agonist of conventional NMDARs4, 6."

If this experiment can not be done, it is complicated to claim in the abstract for example that: "D-serine's actions were mediated via antagonism of a newly discovered class of glutamatergic N-methyl-D-aspartate receptors that are highly calcium permeable and whose expression at excitatory synapses in the MEA renders neurons in this region especially vulnerable to excitotoxicity."

Finally, I have a minor comment on the following hypothesis authors mentioned in the discussion: "The glial perspective– given that normal homeostatic functions of astrocytes are affected by injury, reflected by changes in their reactive status, we propose a switch in their neuroprotective role whereby they are no longer a source of D/L-serine in the MEA during TLE. Indeed, L-serine may be converted by the enzyme serine racemase to pyruvate (via a β -elimination pathway) instead of D-serine (via a racemization pathway) in reactive astrocytes to meet the high metabolic demand of attending to dead and dying cells during seizure activity and/or in neurons attending to their hyperexcitable state."

This is an elegant hypothesis but only hypothesis as to my mind it has never been shown.

RESPONSE TO REVIEWERS' CONCERNS & MODIFICATIONS MADE TO THE MANUSCRIPT (Round 3)

REVIEWER COMMENTS

Reviewer #2 (Remarks to the Author):

First of all, I would like to thank Dr Kumar and his collaborators to have taken into account my previous comments.

However, I still have a problem with the second major point I mentioned last time. It was the following:

2) The second conclusion authors made is regarding the antagonism action of D-serine on t-NMDARs. First of all, authors should show that t-NMDAR activity is higher and d-NMDAR activity is smaller after an epileptogenic status. If this is true this should be rescued by D-serine treatment. Authors have to perform additional electrophysiological experiments to make this strong conclusion.

Authors answer:

“Electrophysiological evidence suggests that ~80% of neurons in the MEA express t-NMDARs (assayed through multiple approaches including molecular biology and characterization of current-voltage relationship using electrophysiology) and we are not quite sure how the conclusion that epileptogenic activity can alter this phenotype was reached. These findings have been published (Kindly see Beesley S et al. (2019) J Neurophysiol. 121: 238-254; Pilli J, Kumar SS (2012) Neuroscience 222:75-88).”

It is clear that D-serine level is decreased in epileptic condition and that D-serine administration has a positive effect. However, authors didn't show that this positive D-serine effect is going through antagonism of t-NMDARs. I understand that it can be long to perform electrophysiological experiments. But, if this is true (antagonism of tNMDARs), D-serine effect should be reproduced by D-AP5 application. Indeed, authors mentioned in the introduction: “These receptors are blocked by the pan-NMDAR antagonist D-(-)-2-Amino-5-phosphonopentanoic acid (D-AP5) and by D-serine, a potential gliotransmitter and a co-agonist of conventional NMDARs^{4, 6}.”

If this experiment can not be done, it is complicated to claim in the abstract for example that: “D-serine's actions were mediated via antagonism of a newly discovered class of glutamatergic N-methyl-D-aspartate receptors that are highly calcium permeable and whose expression at excitatory synapses in the MEA renders neurons in this region especially vulnerable to excitotoxicity.”

>>

1) We believe data pertaining to antagonism of t-NMDARs by D-serine is presented in the manuscript and appears to have been overlooked! Kindly refer to the boxed image in original supplementary figure 1 that shows the differential antagonist effects of D-serine on t- versus d-NMDARs. Previous publications reporting D-serine's effects were also cited in the manuscript (references 4 and 6, Page 2, Paragraph 2).

2) The effect of D-AP5, the pan-NMDAR antagonist, on t-NMDARs has been reported previously in acute brain slices experiments in the ERC (see figure below) and in the somatosensory cortex (Beesley S et al., J Neurophysiol 121, 238-254 (2019); Pilli J, Kumar SS, Neuroscience 222, 75-88 (2012)) and we have **now** included that data here- Please see new supplementary figure 1. We have all the D-AP5 data but chose not to show traces where the EPSCs are completely blocked by the drug. Systemic administration of D-AP5 has deleterious effects including serious neurotoxicity (see J W Olney et al., Science 254 (5037):1515-8 (1991) in animals and hence is not recommended.

3) We don't understand what "higher t-NMDAR activity" means or how it could be measured especially if neurons are dead after status epilepticus and the only way that D-serine can exert its effect is through NMDARs.

4) Higher calcium permeability through t-NMDARs has been reported in the literature and referenced in our manuscript (references 4 and 6). Kindly also see Pilli J, Kumar SS Neuroscience 272:271-285 (2014). We have assessed Ca^{2+} permeability through receptors in the ERC as well using ion exchange experiments but this is beyond the scope of the current work. Kindly note that neurons are lost in these animals on account of seizure activity and D-serine rescues this loss. This couldn't be through d-NMDARs for which it is a known agonist. Our data are consistent with D-serine's demonstrated role as an antagonist for t-NMDARs (see Beesley S et al., J Neurophysiol 121, 238-254 (2019)).

5) The abstract has now been revised to state more precisely the type of NMDARs being assayed in this work and to reflect on the fact that they are not "new" anymore (2012).

Finally, I have a minor comment on the following hypothesis authors mentioned in the discussion: "The glial perspective— given that normal homeostatic functions of astrocytes are affected by injury, reflected by changes in their reactive status, we propose a switch in their neuroprotective role whereby they are no longer a source of D/L-serine in the MEA during TLE. Indeed, L-serine may be converted by the enzyme serine racemase to pyruvate (via a β -elimination pathway) instead of D-serine (via a racemization pathway) in reactive astrocytes to meet the high metabolic demand of attending to dead and dying cells during seizure activity and/or in neurons attending to their hyperexcitable state."

This is an elegant hypothesis but only hypothesis as to my mind it has never been shown.

>> This is after all a hypothesis. Are we not allowed to raise a testable hypothesis through our work? Isn't this how scientific progress made? We have now reworded this as: "The glial perspective— given that normal homeostatic functions of astrocytes are affected by injury, reflected by changes in their reactive status, we propose the hypothesis of a switch in their neuroprotective role whereby they are no longer a source of D/L-serine in the MEA during TLE."

REVIEWER COMMENTS

Reviewer #2 (Remarks to the Author):

Dear authors,

I would like to insist on the fact that I believe in the key important message of this paper. D-serine level is decreased and participate to neuronal loss. Targeting this pathway could have a strong potential impact as a therapeutic agent.

However, as both types of NMDARs are expressed in the MEA, I think that it just too strong to claim the following: "D-serine's actions were mediated via antagonism of glutamatergic GluN3-containing triheteromeric N-methyl-D-aspartate receptors". D-serine could have an action on both and the rescue by D-serine could be due to a rebalance of their respective function.

With kind regards.

REVIEWER COMMENTS

Reviewer #2 (Remarks to the Author):

Dear authors,

I would like to insist on the fact that I believe in the key important message of this paper. D-serine level is decreased and participate to neuronal loss. Targeting this pathway could have a strong potential impact as a therapeutic agent.

However, as both types of NMDARs are expressed in the MEA, I think **that it just too strong** to claim the following: **"D-serine's actions were mediated via antagonism of glutamatergic GluN3-containing triheteromeric N-methyl-D-aspartate receptors"**. **D-serine could have an action on both and the rescue by D-serine could be due to a rebalance of their respective function.**

With kind regards.

>> >> This sentence has now been toned down and reworded as: "D-serine's actions were most likely mediated via antagonism of glutamatergic GluN3-containing triheteromeric N-methyl-D-aspartate receptors"

We appreciate the reviewer's concern and apologize for not being clear enough on this. D-serine does have an action on both GluN3-containing and GluN2-containing NMDARs. For the GluN2-containing NMDARs, D-serine serves as an agonist⁵, which means its infusion, would be pro-convulsive. However, infusion of D-serine does not cause epilepsy; as a matter of fact, it counters the effects of pilocarpine, the convulsant, used in our model to induce status epilepticus. Even in acute slices, D-serine causes no increases in EPSC amplitude (Supplementary Fig. 1). Given that a large majority of neurons (~80%) in the MEA are GluN3 positive³ and if D-serine were to exert its effects through these receptors, it would be through their blockade as an antagonist, and not as an agonist. We have shown in two separate brain regions, the entorhinal cortex^{2,3} and the somatosensory cortex^{1,4} that D-serine blocks t-NMDAR mediated EPSCs just like D-AP5 does (also shown in the supplementary Fig 1 in this manuscript). Hence the only logical conclusion that can be drawn from these observations is that the anticonvulsive effects of D-serine in the MEA are mediated through antagonism of GluN3-containing NMDARs. D-serine's effect through non-GluN3 containing NMDARs are at best negligible because 1) the minimal expression of GluN3-lacking neurons in MEA and 2) the fact that we see neither seizures nor cell loss in the D-serine only control animals (Fig. 1). Additionally, we do not have any reason to believe that infusion of D-serine allows neurons to proliferate to compensate for cells loss.

By "rescue by D-serine could be due to a rebalance of their respective function," is the reviewer implying that there is a redistribution of GluN3 containing t-NMDARs and GluN2 containing d-NMDARs in the epileptic animals? Even if we assume this scenario, we don't understand how D-serine would rescue neurons because it is an agonist for d-NMDARs. Is

the reviewer implying that D-serine's actions somehow alter NMDAR function to reduce Ca^{2+} induced excitotoxicity? Even if this were the scenario, although we do not have a shred of evidence for this, there would be a loss and not a gain or rescue of neurons as is seen in the midmost portions of MEA in this study that have more conventional NMDARs. Hence, the only parsimonious explanation for the observed effects is that D-serine blocks the highly Ca^{2+} permeable t-NMDARs receptors thereby rescuing the loss of the neurons in the MEA during TLE.

1. Pilli J, Kumar SS. Triheteromeric N-methyl-D-aspartate receptors differentiate synaptic inputs onto pyramidal neurons in somatosensory cortex: involvement of the GluN3A subunit. *Neuroscience* 222, 75-88 (2012).
2. Kumar SS. Functional detection of novel triheteromeric NMDA receptors. In: *Ionotropic Glutamate Receptor Technologies* (ed Popescu GK). Springer (2016).
3. Beesley S, Sullenberger T, Pilli J, Abbasi S, Gunjan A, Kumar SS. Colocalization of distinct NMDA receptor subtypes at excitatory synapses in the entorhinal cortex. *J Neurophysiol* 121, 238-254 (2019).
4. Pilli J, Kumar SS. Potentiation of convergent synaptic inputs onto pyramidal neurons in somatosensory cortex: dependence on brain wave frequencies and NMDA receptor subunit composition. *Neuroscience* 272, 271-285 (2014).
5. Mothet JP, et al. D-serine is an endogenous ligand for the glycine site of the N-methyl-D-aspartate receptor. *Proc Natl Acad Sci U S A* 97, 4926-4931 (2000).